# CoordX: Accelerating Implicit Neural Representation with a Split MLP Architecture

**Ruofan Liang[1,2], Hongyi Sun[1], Nandita Vijaykumar[1,2]**
[1]Department of Computer Science, University of Toronto
[2]Vector Institute, Canada
`{ruofan,nandita}@cs.toronto.edu, hongyi.sun@mail.utoronto.ca`

## Abstract

Implicit neural representations with multi-layer perceptrons (MLPs) have recently gained prominence for a wide variety of tasks such as novel view synthesis and 3D object representation and rendering. However, a significant challenge with these representations is that both training and inference with an MLP over a large number of input coordinates to learn and represent an image, video, or 3D object, require large amounts of computation and incur long processing times. In this work, we aim to accelerate inference and training of coordinate-based MLPs for implicit neural representations by proposing a new split MLP architecture, *CoordX*. With CoordX, the initial layers are *split* to learn each dimension of the input coordinates separately. The intermediate features are then fused by the last layers to generate the learned signal at the corresponding coordinate point. This significantly reduces the amount of computation required and leads to large speedups in training and inference, while achieving similar accuracy as the baseline MLP. This approach thus aims at first learning functions that are a decomposition of the original signal and then fusing them to generate the learned signal. Our proposed architecture can be generally used for many implicit neural representation tasks with no additional memory overheads. We demonstrate a speedup of up to 2.92x compared to the baseline model for image, video, and 3D shape representation and rendering tasks.

## 1 Introduction

Traditional discrete representations, including volumes or point clouds, have long been used for tasks such as 3D object representation and novel view synthesis. Until recently, implicit neural representations have emerged as a popular alternative approach to those tasks. These *implicit neural representations* leverage multilayer perceptrons (MLPs) to represent images, videos, 3D objects, etc (Mildenhall et al., 2020; Sitzmann et al., 2020; Tancik et al., 2020). These MLPs (coordinate-based MLPs, coordMLP for short) take low dimensional coordinates (e.g., coordinate position or time index) as input and predict the value of a learned signal (e.g., color, depth, shape, etc.) of each coordinate point. The coordinate-based MLP learns an implicit, continuous, and differentiable function that maps input coordinates to various signals.

There are several benefits to using an implicit neural representation instead of a traditional discrete representation. First, it is potentially more memory efficient. Unlike the discrete representation where the granularity of the represented object is limited by the grid resolution, the granularity of an implicit neural representation can be freely adjusted by choosing different resolutions of input coordinates while not exceeding the representation capacity of the learned MLP. Second, the fully differentiable learning process allows the MLP to learn complex signals from *sparsely* available data points to reconstruct high-quality images or objects. Recent research has demonstrated the effectiveness of coordMLPs for a wide range of signal fitting or learning tasks such as image super-resolution (Chen et al., 2021), 3D shape representation (Park et al., 2019; Mescheder et al., 2019; Gropp et al., 2020), novel view synthesis (Mildenhall et al., 2020; Martin-Brualla et al., 2021; Schwarz et al., 2020) and photo-realistic 3D scene editing (Niemeyer & Geiger, 2021). The search for more accurate and generalizable implicit neural network architectures and methodologies is an active area of research.

However, a key challenge of implicit neural representation is the significant computation demand for both inference and training. For inference, the MLP has to be queried with *each* positional point to generate the learned signal at the corresponding position. Using one GTX1080 GPU, rendering a $1024 \times 1024$ sized image on 5-layer coordMLP takes 0.2 seconds, and rendering an image of size $400 \times 400$ from a novel view with the NeRF model takes 18 seconds. Both tasks perform millions of MLP queries. Similarly, training an MLP for neural representation is a very computation-intensive task as each MLP requires full training passes over a large number of sample points from the original signal. On a high-end RTX GPU, it takes more than 10 minutes to learn a $512 \times 512$ image using a 5-layer coordMLP and more than 4 hours to reconstruct a low-resolution scene with NeRF.

Several recent works aim to improve the speed of inference and/or training for coordinate-based MLPs for novel view synthesis tasks (Liu et al., 2020; Neff et al., 2021; Garbin et al., 2021; Yu et al., 2021) or object representation (Takikawa et al., 2021). These works leverage spatial sparsity inherent in images, videos, and objects to reduce the amount of computation required for inference or training. For example, NSVF (Liu et al., 2020) constructs a sparse discrete representation using a voxel octree. An implicit neural representation is then learned over the discrete set of voxels instead of the infinite set of coordinates. However, there are several drawbacks to these approaches. First, some of the mentioned approaches are only applicable to specific tasks, e.g., novel view synthesis (Neff et al., 2021). Second, the additional data structures that are used to explicitly save representations to avoid computation may lead to high memory overheads, e.g., Garbin et al. (2021). Third, the effectiveness of these approaches is limited by the sparsity of the signal being learned and they may be less effective for highly dense and feature-rich images/objects.

Our goal, in this work, is to accelerate both training and inference for a wide range of coordMLP-based representation tasks. To this end, we leverage the following observation. The inputs to these MLPs are typically multi-dimensional coordinate points, i.e., an $(x, y)$ coordinate for an image, $(x, y, t)$ for a video, etc., and in existing MLPs, each coordinate is processed independently, as depicted in Fig. 1(d). However, this misses the implicit locality among points in an image that have similar coordinate values, for example, the neighboring points along the same axis. In this work, we propose a new architecture where the initial layers are *split* to independently process each dimension of the input. Thus, each *dimension* is treated independently rather than each coordinate point itself. The outputs of these initial layers are then fused to generate a final prediction based on all input dimensions, as depicted in Fig. 1(e). We refer to this split architecture as *CoordX*.

The major benefits of our approach are: First, it enables significant speedups in both training and inference by essentially reducing the dimensions of the input. For example, the overall inputs of an $H \times W$ image fed into the MLP is reduced from $H \times W$ to $H + W$. At the same time, by leveraging the locality between points with similar coordinate values, we are able to achieve accuracy close to the baseline architecture. Second, this technique can be applied orthogonally to previously-proposed optimization techniques that leverage sparsity without incurring much additional memory overheads. Third, the dimension-based splitting approach can be applied to a wide range of tasks that use coordinate-based MLPs as we demonstrate in this work.

The key challenges in developing a split architecture are: (i) ensuring that the overall size of the model does not increase even when processing each dimension of the input with a separate layer; and (ii) effectively fusing features from the split layers to retain the original model accuracy. In this work, we handle these challenges by (i) splitting only the first FC layer and then sharing the remaining FC layers across all input branches; and (ii) fusing features from split branches using an outer product operation. We demonstrate that CoordX can significantly accelerate training and inference for various signal fitting tasks[1] including image, video, and 3D shape fitting. Our analysis shows a speedup of ∼2.5x using a 5-layer CoordX model for various tasks. We demonstrate a 2.35x speedup in inference time for a 1k resolution image and a 2.78x speedup for a 1k resolution video clip in our experiments.

To summarize, in this work, we make the following contributions:

- We develop a new architecture for coordinate-based MLPs that leverages locality between input coordinate points to reduce the amount of computation required and significantly speed up inference and training.

---

[1]CoordX however, does not improve the performance when fitting unidimensional signals (e.g., audio).

- We design the split MLP architecture that achieves faster training and inference and retains the same level of accuracy as the baseline models with comparable or slightly larger parameter counts.
- We demonstrate CoordX's effectiveness in speeding up inference and training on various signal fitting tasks, including image, video and 3D shape rendering, and other advanced applications such as image super-resolution.

## 2 RELATED WORK

**Coordinate-based MLPs for implicit neural representation.** Recent works have demonstrated the potential of MLPs in implicitly representing signals such as images (Stanley, 2007), volume occupancy (Peng et al., 2020; Mescheder et al., 2019), signed distance (SDF) (Park et al., 2019; Xu et al., 2019), texture (Saito et al., 2019; Henzler et al., 2020), and radiance fields (NeRF) (Mildenhall et al., 2020). Unlike MLPs used for other tasks, an MLP used for implicit representation needs to be able to capture the high frequency variations in the color or geometry of images and objects. To address this, Mildenhall et al. (2020) proposes a *positional encoding* (PE) method that maps the original input coordinates into a high dimensional space using a series of high frequency functions to better capture high frequency variations. Sitzmann et al. (2020) use *sinusoidal activation* functions (SIREN) to replace the ReLU activations in the MLP, and Tancik et al. (2020) use simple Fourier feature mappings of input coordinates to learn high frequency variations. Many recent works also aim to make the signals represented by coordinate-based MLPs editable or generalizable. A common way to enrich the representation power of these models is to add latent codes/features as additional inputs to the coordMLPs (Schwarz et al., 2020; Park et al., 2021; Chen et al., 2021; Mehta et al., 2021). These additional inputs are combined with positional coordinates and are essentially treated as additional dimensions in the input coordinates. CoordX is directly applicable to these works.

**Accelerating MLPs for implicit neural representation.** Several recent works aim to accelerate inference and/or training for implicit neural representation using MLPs. There are broadly two directions among existing works: (i) reducing the number of points to be queried; (ii) reducing the time spent in processing each point. Reducing the number of query points is a common optimization method for ray-casting-based 3D neural representations. Due to spatial sparsity in a 3D scene, a significant number of sampled points along cast rays are empty, without useful density or color information. The inference/training speed can be greatly improved by skipping the processing of these points. Neff et al. (2021) use an additional depth network to estimate object depth along each ray, hence only points around the object surface are sampled. Similarly, traditional explicit 3D representation methods can be leveraged to guide point sampling. Garbin et al. (2021) and Kellnhofer et al. (2021) explicitly extract the mesh of a rendered object after training to reduce the number of points that need to be processed by the MLP. Liu et al. (2020); Yu et al. (2021); Takikawa et al. (2021) construct an octree structure to represent the 3D object during training that enables skipping empty spaces. To reduce the computation done for each point during inference, Reiser et al. (2021) replace the larger MLP for the whole scene with many tiny MLPs that have fewer layers and parameters, each of which only represents a small region of the whole scene. Garbin et al. (2021) and Yu et al. (2021) also pre-compute points and store results in memory. Therefore, only lightweight interpolation rather than intensive MLP computation is performed for each query point during rendering. To alleviate the significant memory requirements and processing time to pre-compute the original NeRF MLP's 5D input grid (3D position + 2D direction), these works leverage spherical harmonics (Ramamoorthi & Hanrahan, 2001) to generate a more efficient 3D representation.

Concurrent work ACORN (Martel et al., 2021) uses a multi-scale block-coordinate decomposition to efficiently represent large-scale images or 3D shapes in a hierarchical way. Unlike our coordinate decomposition, ACORN uses a quadtree or octree to decompose the input coordinate grid into small patches. ACORN first gets patch-level features via a large coordinate-based MLP, and then uses these features to get predicted signals for each coordinate via feature interpolation and a lightweight MLP. Thus the number of queries to the large coordMLP in ACORN is significantly reduced. Though both ACORN and CoordX reduce the number of queries to some parts of the coordinate-based model, they use very different approaches and these two approaches are orthogonal. CoordX can be combined with ACORN to generate additional speedups by replacing ACORN's coordinate encoder with the CoordX model.

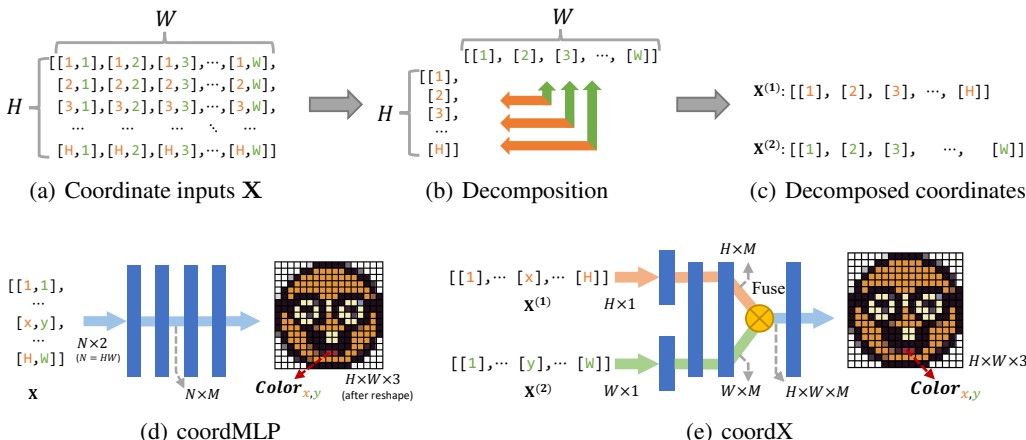

Figure 1: (a)-(c) The process of decomposing coordinate inputs $\mathbf{X}$ for 2D image of size $H \times W$ into 2 split parts $\mathbf{X}^{(1)}$ and $\mathbf{X}^{(2)}$. (d) Baseline coordMLP architecture. $N = HW$ coordinate inputs ($\mathbf{X}$) are independently fed into the MLP. (e) Our CoordX architecture. The initial layer is split up into separate branches for each decomposed coordinates $\mathbf{X}^{(1)}$ and $\mathbf{X}^{(2)}$. After shared FC layers, the split features are fused and the fused features are then fed into a few FC layers to generate the final signal values. $M$ is the size of the hidden features.

## 3 METHOD

### 3.1 NEURAL NETWORK ARCHITECTURE

In our proposed architecture, the initial layer of the MLP is split across $C$ input branches. The following $D_s - 1$ FC layers are shared among the input branches. The fusion operation then fuses the intermediate features from the different feature branches and feeds fused features to last $D_f$ FC layers to generate the final signal values (see Fig. 1(e)).

**Input decomposition along coordinate dimensions.** In typical coordMLP models, a set of $N$ coordinate points are individually queried as inputs, where $N = HW$ for an image of size $H \times W$ in Fig. 1(d). We refer to this set of input coordinate points as $\mathbf{X} \in \mathbb{R}^{N \times K}$, where $K$ is the dimension of each point (e.g., $K = 2$ for $(x, y)$ points on a pixel plane). With CoordX, instead of querying the first layer of the MLP with $N$ coordinate points of dimension $K$, we split the first layer of the MLP into $C$ separate branches, where $C \leq K$. If $K = C$, each dimension $i$ of the $K$-D input coordinate is separately fed to the corresponding $C_i$ input branch as shown in Fig 1(e). If $C < K$, in this case, 1 or more coordinate dimensions are not split and retained their original form. For example, coordinate points in a video $(x, y, t)$ ($K = 3$) can be decomposed along 2 dimensions ($C = 2$): branch 1 is fed by 2D coordinate points along the X and Y dimensions as $(x, y)$ and branch 2 is fed by the temporal dimension ($t$).

Thus the input grid $\mathbf{X} \in \mathbb{R}^{N \times K}$ is transformed into a set of decomposed input grids $\{\mathbf{X}^{(i)} \in \mathbb{R}^{B_i \times K_i} \mid i = 1, 2, \ldots, C\}$, where $i$ is the input branch, $B_i$ is the number of coordinate points fed into branch $i$ and $K_i$ is the dimension of the each decomposed coordinate point fed into branch $i$. Thus, $N = \prod_{i=1}^{C} B_i$ and $K = \sum_{i=1}^{C} K_i$.

**First layer.** The first layer of CoordX contains $C$ parallel branches converting $\mathbf{X}^{(i)}$ to the corresponding hidden features $\mathbf{H}_1^{(i)}$. Each branch $i$ has a weight matrix $\mathbf{W}_1^{(i)} \in \mathbb{R}^{K_i \times M}$, where $M$ is output feature size. With activation function $\phi(\cdot)$, the first layer is [2]:

$$\{\mathbf{H}_1^{(i)} = \phi(\mathbf{W}_1^{(i)} \mathbf{X}^{(i)}) \mid i = 1, 2, \ldots, C\} \tag{1}$$

**Layers before fusion.** To avoid adding additional parameters to our model, we use the shared weight matrices (FC layers) to process hidden features from all parallel branches after the first layer (Fig. 1(e)). Suppose there are $D_s$ layers before fusion, we can formulate these layers as:

$$\{\mathbf{H}_j^{(i)} = \phi(\mathbf{W}_j \mathbf{H}_{j-1}^{(i)}) \mid i = 1, 2, \ldots, C\}, \text{for } j = 2, \ldots, D_s \tag{2}$$

---

[2]We omit the bias term $\mathbf{b}$ in the following equations for readability.

This design has the following benefits: (i) no extra parameters are added because the first layer is split among the branches and the remaining layers are shared; (ii) higher computational efficiency/parallelism as features from different branches can be concatenated and processed in one batch; (iii) sharing parameters between input branches enables capturing correlations between the different input dimensions.

**Feature Fusion.** The intermediate features $\mathbf{H}_{D_s}^{(i)} \in \mathbb{R}^{B_i \times M}$ from the $C$ branches are then combined to generate a fused feature $\mathbf{H}_{D_s}^{*} \in \mathbb{R}^{B_1 \times \cdots \times B_C \times M}$.

We use an outer product as the fusion operation (discussed in more detail in 3.3). The fusion operation is formulated as follows:

$$\mathbf{H}_{D_s}^{*} = \text{Fuse}(\mathbf{H}_{D_s}^{(1)}, \cdots, \mathbf{H}_{D_s}^{(C)}) = \mathbf{H}_{D_s}^{(1)} \otimes \cdots \otimes \mathbf{H}_{D_s}^{(C)} \tag{3}$$

Where $\otimes$ denotes the outer product operation. The last $D_f$ FC layers from the baseline model are retained to transform the fused features into final predictions.

**Speedup Analysis.** We assume each FC layer of the coordinate-based MLP has the same amount of computation. In a $D$-layer baseline coordMLP, there are $N$ coordinate inputs processed by each layer of the MLP. The resulting computation is of the order $\mathcal{O}(DN)$. In a $D$-layer CoordX model with $C$ split branches, we assume $B_i$ (the number points processed by branch $i$) equals $B$, thus $N = \prod_{i=1}^{C} B_i = B^C$. Each of the $C$ branches from the first $D_s$ layers before fusion needs to process $B$ decomposed inputs and the remaining $D_f$ layers ($D_s + D_f = D$) process the fused features of the original size $N$. The two parts in CoordX together result in computation of the order $\mathcal{O}(CD_sB + D_fN)$. The ratio $\gamma$ of the computational complexity between the two methods is

$$\gamma = \frac{DN}{CD_sB + D_fN} = \frac{(D_s + D_f)B^C}{CD_sB + D_fB^C} = \frac{\lambda + 1}{\lambda CB^{1-C} + 1}, \quad \lambda = \frac{D_s}{D_f} \tag{4}$$

When $B$ is large enough, the term $\lambda CB^{1-C}$ becomes negligibly small. Thus, the theoretical upper bound of the speedup that can be achieved is $\gamma \to \lambda + 1$. A 5-layer CoordX MLP with $D_s = 3$ and $D_f = 2$ can thus achieve a 2.5x speedup over a 5-layer baseline MLP based on our assumptions.

## 3.2 Accelerating Training with CoordX

When input coordinates are split along each coordinate dimension, during fusion, the original coordinates are reconstructed by generating all combinations of the input values along each dimension. Thus, to perform this splitting for training, the original signal values must be available for all combinations of the split inputs. For example, if the pixel points $[(1,1), (2,2), (3,3), (4,4)]$ are sampled, they cannot be split into two input vectors for each coordinate dimension, because some combinations of the input vectors are missing (e.g., $(1,3), (1,4), \dots$). If points $[(1,1), (1,2), (2,1), (2,2)]$ are sampled, we can then split them into two $[(1), (2)]$ vectors along each dimension. In a typical training process, points are randomly sampled and all combinations will not be available.

To address this, we first sample positions along each decomposed dimension, then use these sampled positions to generate all combinations of coordinate points (see Fig. 4). The number of sampled positions along each dimension is approximately proportional to the dimension length so that each separate branch has a sufficient number of decomposed coordinates to learn while the uniform distribution of the sampled points is still guaranteed. We discuss how this is done in more detail in Appendix A.

## 3.3 The signal decomposition of CoordX.

Signals such as images can be decomposed into multiple matrices using mathematical methods. For example, a gray-scale image (represented as a 2D matrix) can be decomposed into two matrices using an SVD or QR decomposition. Similarly, our proposed split architecture can essentially be thought of as learning multiple matrices that are then used to reconstruct the original signal, as depicted in Fig. 2. By moving the fusion operation (outer product) to the end of the model ($D_f = 0$), we get a fully split CoordX model. This fully split CoordX model is also able to represent the image with high PSNR (Fig. 2(b)), indicating the ability of the model to learn decomposed signals.

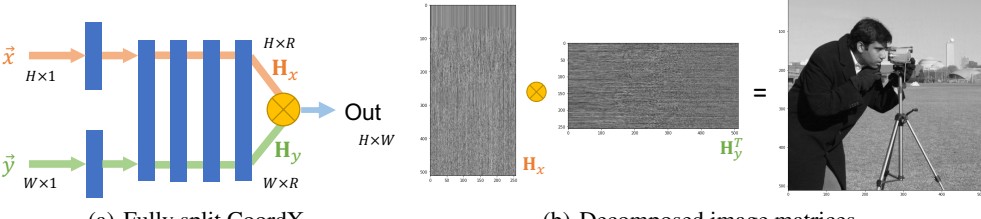

(a) Fully split CoordX        (b) Decomposed image matrices

Figure 2: CoordX learns the image decomposition. (a) The fully split coordX architecture. The fusion operator directly generates the predicted signal. $R$ is the dimension reduced by the summation after the outer product. (b) The image can be represented as a matrix multiplication between two learned feature tensors (matrices) $\mathbf{H}_x$ and $\mathbf{H}_y$. $R = 256$, $H = W = 512$ in this example.

A key difference between the previously-presented CoordX model (e.g., Fig. 1(e)) and the fully split CoordX model (e.g., Fig. 2(a)) is the location of the fusion operation. Our fusion operation defined in Eqn. 13 is a series of outer products, while the fully split CoordX in Fig. 2 requires an outer product to generate the final prediction without an additional FC layer. In order to include these two cases in one fusion equation, we extend the dimension of the split features to be fused. Let the hidden features after $D_s$ layers are of feature size $M$. The $M$ hidden feature values then can be reshaped into a $R \times S$ matrix (where $M = RS$). $R$ is the size of the dimension reduced by the summation after outer product, and $S$ represents the feature/signal size after the fusion. For example, $R = 1, S = M$ for fusion in the partially-split CoordX. The fully split CoordX in Fig. 2 has $R = 256$ and $S = 1$ (single channel image). Following our definition in 3.1, each branch's hidden feature $\mathbf{H}^{(i)} \in \mathbb{R}^{B_i \times M}$ before the fusion can be reshaped into $\mathbf{H}'^{(i)} \in \mathbb{R}^{B_i \times R \times S}$. The fusion then can be formulated as

$$\mathbf{H}^*_{p_1 p_2 \cdots p_C} = \sum_{i=1}^{R} (\prod_{j=1}^{C} \mathbf{H}'^{(j)}_{p_j, i}) \tag{5}$$

$p_i$ is positional index for feature tensors, $i$ is the index for dimension $R$ of $\mathbf{H}'^{(i)}$, $\mathbf{H} \in \mathbb{R}^{B_1 \times \cdots \times B_C \times S}$. The fusion operation defined in Eqn. 13 is a special case of Eqn. 5 when $R = 1$. While $R = 1$ is sufficient to learn an accurate representation for most signals, if we set $R > 1$ (e.g., $R$ is 2 or 3) for the hidden feature fusion and enlarge the corresponding FC layers by a scale factor $R$ to keep $S$ unchanged, the representation accuracy of the CoordX model can be improved (evaluated in Sec. 4.4).

## 4 EXPERIMENTAL EVALUATION

In this section, we evaluate the performance and efficiency of CoordX for several signal fitting tasks, including images, videos and 3D shapes. We also evaluate CoordX for latent code coordMLPs.

**Setup.** We choose several existing coordMLP models as baseline models for different tasks (details are in the following subsections). All models are implemented on PyTorch (Paszke et al., 2019) and evaluated using an NVIDIA RTX3090 GPU. We use SIREN to denote models with periodic activation functions (Sitzmann et al., 2020), and we use PE to denote models using positional encoding with ReLU activation (Mildenhall et al., 2020; Tancik et al., 2020). Unless otherwise specified, our experiments use 5-layer MLPs. The CoordX models have two FC layers after the fusion operation ($D_f = 2$). Model hyperparameters such as learning rate, batch size, number of epochs, etc., are the same as those used in the corresponding baseline coordMLPs. The output image/object visualizations and qualitative comparisons are in Appendix B.

### 4.1 SIGNAL FITTING WITH COORDX

**Image Representation.** In this experiment, 5-layer coordMLPs with 256 hidden units are used to fit RGB images of size $512 \times 512$. We randomly select 12 center-cropped images from DIV2K dataset (Agustsson & Timofte, 2017) to report the average PSNR. Both SIREN and PE models are tested. Table 1 shows the PSNR and the required training time for 20,000 epochs. Fig. 3(a) compares the inference speed between the baseline SIREN model and CoordX for different image sizes.

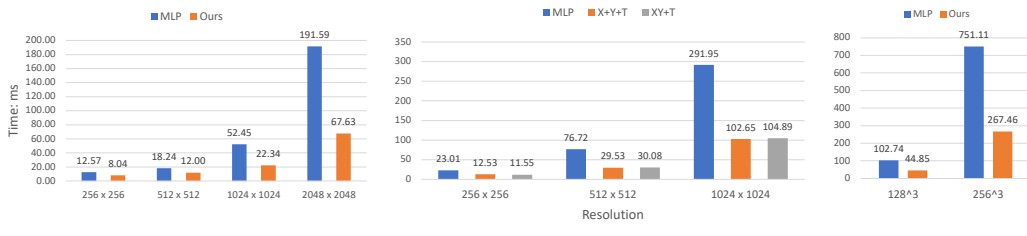

|              | (a) Image inference | (b) Video inference | (c) Shape inference |

Figure 3: Comparing inference speed. (a) average inference time for the SIREN model for a synthetic image on different resolutions; (b) millisecond per frame to represent a synthetic video with 100 frames on different resolutions with PE models; (c) inference time of a grid of coordinate points for marching cubes to display 3D shapes; SIREN models are used.

| Method | MLP-S | CoordX-S | CoordX-S-A | MLP-P | CoordX-P | CoordX-P-A |
|---|---|---|---|---|---|---|
| PSNR | 40.25 | 37.08 | 37.02 | 37.30 | 33.89 | 33.84 |
| $T_{\text{training}}$ (min) | 12.07 | 22.33 | 5.73 | 14.80 | 27.08 | 5.06 |

Table 1: Accuracy and speed comparison. MLP indicates the baseline coordMLP. S and P refer to SIREN and PE respectively. A refers to accelerated training with our sampling method (Sec. 3.2).

We make the following observations. First, CoordX can achieve high PSNR values similar to those achieved by the corresponding baseline coordMLPs, with a ∼3 dB drop in PSNR (we discuss ways to alleviate this in 4.4). Second, training the CoordX model is almost twice as fast as training the baseline coordMLP. Third, CoordX achieves significantly faster inference times compared to the baseline (up to 2.83x for a 2K image). The observed speedups are higher for bigger images.

**Video Representation.** In this task, we use 5-layer coordMLPs with 1024 hidden units to learn 10-second video clips. We evaluate 2 video clips. The baseline MLPs are described in Appendix B.

For the CoordX model, we consider two splitting strategies. The first one is to split the input coordinates into 3 vectors along the height, width, and time axes, resulting in 3 network branches (referred to as X+Y+T). The second strategy splits the input coordinates along the spatial and temporal dimensions (2 branches). The height and width dimensions are flattened into one input branch and the time dimension is the second branch. We refer to this strategy as XY+T. We use the PE method in this experiment as it performs better than SIREN for videos. Table 2 shows the average PSNR and the required training time for 100,000 epochs. Fig. 3(b) compares the inference speed between the baseline model and CoordX for different frame resolutions.

We make the following observations. First, while both X+Y+T and XY+T splitting strategies achieve high PSNR, XY+T performs slightly better than X+Y+T and even outperforms the baseline model. Second, training CoordX with our sampling strategy (XY+T-A) is 2.02X faster than training the baseline model while retaining high representation quality in the model. Third, CoordX achieves a 2.78x speedup over the baseline MLP for inference, and the speedups are higher for higher video resolutions.

**3D shape representation.** We choose occupancy networks (Mescheder et al., 2019) to represent 3D shapes. The output probabilities indicate whether query points are inside or outside the shape. We use 5-layer coordMLPs with 256 hidden units for this experiment and the SIREN method as it performs better than PE.

We use IoU as the quality metric (see details in Appendix B) and report the average IoU obtained for 4 3D objects. Table 2 shows the average IoU values and the required training time for 10,000 epochs. To display shapes and compare inference speeds after training, we use the marching cube[3] (Lorensen & Cline, 1987) process in which a 3D voxel grid needs to be queried. Fig. 3(c) compares the inference speed for different grid resolutions.

We make the following observations. First, we achieve similar IoU scores with CoordX compared to the baseline coordMLPs. Second, CoordX is up to 1.7x faster for training speedup compared to the baseline model. Third, CoordX achieve a speedup of up to 2.81x over the baseline for a resolution of $256^3$ and on average 2.55x.

---

[3]Volume rendering, as another way to display 3D shapes, will be discussed in Appendix C.

| Task | Video | | | | 3D shape | | |
|------|-------|------|------|--------|----------|--------|---------|
| Method | MLP | X+Y+T | XY+T | XY+T-A | MLP | CoordX | CoordX-A |
| Metric | 33.315 | 31.69 | 33.63 | 33.35 | 0.991 / 0.970 | 0.991 / 0.970 | 0.989 / 0.964 |
| $T_{\text{training}}$ | 4.30 h | 6.63 h | 5.22 h | 2.13 h | 5.72 min | 16.14 min | 3.28 min |

Table 2: Accuracy and speed comparison for MLP training. MLP indicates the baseline coordMLP. $A$ refers to training CoordX with our sampling strategy. Video task uses PSNR as the metric, and 3D shape task uses easy / hard IoU (higher is better) as metrics.

In summary, we demonstrate that the CoordX models are able to provide significant speedups in both training and inference over baseline models for different sets of tasks. We also demonstrate that the CoordX models are able to retain the representation quality of the baseline models.

**Novel View Synthesis**. CoordX can also be used to accelerate NeRF-like volume rendering (Milden-hall et al., 2020). However, NeRF has an irregular input coordinate grid (see Fig. 8), and thus CoordX cannot be directly applied here. We instead use a simple mechanism in C.4 to accelerate grid-interpolation-based NeRF rendering with CoordX. This mechanism involves the precomputa-tion of densely sampled coordinate points and grid interpolation over the MLP predictions. Our proposed method can achieve a 1.81x training speedup and a 4.41x rendering speedup over baseline NeRF with a quality drop of less than 0.5 dB on low-resolution blender scenes.

## 4.2 COORDX WITH LATENT CODES

In this part, we test our method on more complex coordinate-based MLPs, which include high-dimensional latent codes as a part of the model input for better generalizability. The model we accelerate is LIIF (Local Implicit Image Function) (Chen et al., 2021) for image super-resolution. It takes a 2D pixel position and a high-dimensional LIIF latent code from a CNN encoder as inputs and then predicts the RGB value at the corresponding pixel coordinate. The coordMLP (decoder) here is a 5-layer MLP with 256 hidden units. We use a CoordX model with 4 branches (code,x,y,cell) to replace the baseline coordMLP for this task, where a cell is an additional 2D input introduced in Chen et al. (2021) for better accuracy. We use a pre-trained EDSR (Lim et al., 2017) decoder and train our CoordX decoder from scratch for 1,000 epochs. We use the DIV2K dataset (Agustsson & Timofte, 2017) for quantitative comparisons. Table 3 compares the baseline LIIF and our modified CoordX LIIF on different up-sampling scales, including scales used in training and not used in training.

We observe that the CoordX LIIF model achieves almost the same PSNR compared to the original baseline model. At the same time, CoordX achieves significant speedups (up to 2.88x) for inference for a target 2K image super-resolution. Since the MLP only forms a part of the overall training pipeline, CoordX does not generate significant speedups for training. We conclude that CoordX is more generally applicable to variations and enhancements of coordinate-based MLPs.

| Method | In-distribution | | | Out-of-distribution | | | | | Speed |
|--------|-----------------|-----|-----|---------------------|------|------|------|------|-------|
| Up-sampling scales | $\times 2$ | $\times 3$ | $\times 4$ | $\times 6$ | $\times 12$ | $\times 18$ | $\times 24$ | $\times 30$ | ($\times 2$) |
| EDSR-baseline-LIIF | 34.67 | 30.96 | 29.00 | 26.75 | 23.71 | 22.17 | 21.18 | 20.48 | 131.03 ms |
| EDSR-CoordX-LIIF | 34.65 | 30.95 | 28.99 | 26.67 | 23.60 | 22.08 | 21.10 | 20.42 | 45.53 ms |

Table 3: Accuracy and speed comparison for MLP inference. PSNR is used as the quantitative metric. (Scores for EDSR-baseline-LIIF are from the original paper.) The speed here is the average time to finish processing all images in the DIV2K valid set with $\times 2$ scale.

## 4.3 THE DEGREE OF SPLITTING FOR MLP

CoordX models can be partly split or fully split (Sec. 3.3) and in this section, we evaluate the impact of different degrees of splitting for 3D shape fitting. We treat the number of layers after fusion as a hyperparameter ($D_f$) to be swept. We vary $D_f$ from 0 to 5 for the 5-layer CoordX models evaluated. Note that when $D_f = 5$ the coordX model is essentially a baseline coordMLP. To demonstrate that similar representation accuracies as CoordX cannot be obtained by simply using MLPs with fewer layers, we also evaluate baseline coordMLPs with different model depth ($D_{\text{MLP}}$). All CoordX models use our sampling strategy (Sec. 3.2) during training. We use a $128^3$ grid of coordinate points to test the inference speed. The test results are listed in Table 4.

| Model setting | $D_f$ | | | | | | $D_{\mathrm{MLP}}$ | | |
|---|---|---|---|---|---|---|---|---|---|
| | 5* | 4 | 3 | 2 | 1 | 0 | 4 | 3 | 2 |
| IoU | 0.976 | 0.980 | 0.974 | 0.963 | 0.941 | 0.880 | 0.959 | 0.876 | 0.516 |
| $T_{\mathrm{training}}$ (min) | 5.72 | 5.40 | 4.24 | 3.28 | 2.27 | 2.38 | 4.44 | 3.11 | 1.67 |
| $T_{\mathrm{inference}}$ (ms) | 105.22 | 95.43 | 69.37 | 46.25 | 19.73 | 22.18 | 82.12 | 55.21 | 30.77 |

Table 4: Accuracy and speed comparison. When $D_f = 5$, CoordX becomes the baseline coordMLP.

We make the following observations. First, as we have more split layers ($D_f$ is lower), the IoU score decreases slightly. This indicates a tradeoff between representation accuracy and training/inference speed. Second, from our comparison with smaller baseline MLP models, we observe that similar representation accuracies cannot be obtained by simply using fewer layers (which may be faster).

### 4.4 IMPROVING REPRESENTATION QUALITY

As observed in Section 4.1, CoordX leads to significant speedups but causes a drop in accuracy ($\sim$3dB PNSR) specifically for image representation. Compared to other types of signals, image signals typically have more high frequency variations. The decomposition of inputs in CoordX may lead to loss of information in capturing these high frequency signal variations.

To improve the representation quality of CoordX as discussed in Section 3.3, we increase the size $R$ of the reduction dimension ($R = 1$ for models evaluated previously) and keep the fused feature size $S$ unchanged ($S = 256$ for image models). We increase $R$ to 2 and 3 in this evaluation and extra parameters are thus added to the splitting FC layers. This can be done in two ways. First, we can increase the number of hidden units in the layer just before fusion by a factor of $R$ (referred to as +). Alternatively, we can increase the feature size of each splitting layer before fusion by a factor of $R$ (referred to as ++). This approach requires the use of more parameters.

We use SIREN models for this experiment and the experimental settings are the same as in 4.1. Table 5 compares the different model configurations.

| Model | MLP | CoordX | R2+ | R2++ | R3+ | R3++ |
|---|---|---|---|---|---|---|
| PSNR | 40.25 | 37.08 | 39.70 | 41.32 | 41.60 | 43.55 |
| Training time (min) | 12.60 | 5.87 | 7.67 | 7.87 | 9.00 | 9.33 |
| Inference speed (ms) | 52.45 | 22.34 | 29.24 | 34.92 | 30.05 | 37.73 |
| Model size (MB) | 0.77 | 0.77 | 1.02 | 2.27 | 1.27 | 4.78 |

Table 5: Image fitting with larger CoordX models. The inference speed is the average time to inference a synthetic 1024x1024 image 100 times.

From Table 5, we make the following observations. First, increasing $R$ can effectively improve the representation quality. The augmented CoordX models are also able to achieve higher PSNR than the baseline coordMLPs. Second, even though more parameters are added to the augmented CoordX models, the training and inference speeds are still faster than the baseline coordMLP. Third, for the same $R$, the R++ strategy which requires more parameters is more effective. However, increasing the $R$ value is more beneficial compared to using more parameters with the R++ strategy.

## 5 CONCLUSION AND FUTURE WORK

In this work, we propose CoordX, a new architecture accelerating the inference and training of coordinate-based MLPs for implicit neural representations. CoordX decomposes input positional(-temporal) coordinates along each input dimension and splits the baseline MLP into a multi-branch MLP. Decomposed inputs separately feed into each branch leading to significant speedups in inference and training while retaining similar representation accuracy as the slower baseline models. We also demonstrate approaches to increase the representation capacity of a split coordinate-based MLP, leading to new design space for trading off speed, quality and size.

CoordX also has wider applicability beyond the tasks evaluated. For example, the splitting strategy introduced in this work can also be applied to accelerate view synthesis for dynamic scenes (Peng et al., 2021; Li et al., 2021), and deformable scenes (Park et al., 2020; 2021). CoordX can also help neural implicit representation based compression (Dupont et al., 2021) achieve higher compression/decompression speed. Lastly, the split MLPs proposed in this work can potentially be used for multi-dimensional signal decomposition or multivariate function decomposition, which may be useful for other tasks such as multi-relational data modeling (Rabanser et al., 2017). We leave these explorations to future work.

ACKNOWLEDGMENTS

We thank Chen Yang, Zanwei Zhou for interesting discussions throughout the project. We also thank Gavin Guan, Jimmy Lin, Sankeerth Durvasula as well as anonymous reviewers for helpful comments and suggestions of this paper; Bin Ji, Shunyu Yao, Dennis Huang from VokaTech, Shanghai Jiao Tong University for computing support.

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

## A    Input Point Sampling for CoordX Training

Randomly sampling points for MLP training will not lead to any speedups with CoordX. This is because randomly chosen coordinates cannot be decomposed into input vectors along each dimension, as discussed in 3.2.

Thus, we need a new sampling strategy that ensures that the chosen points can be split along each coordinate dimension. To ensure this, we need to select all input points that are combinations of the split input vectors. To this end, we formulate a simple sampling strategy that we describe below.

Suppose we need to sample $N$ points to learn the signal bounded in a $C$-D coordinate space, where each dimension $i$ is of length $S_i$. We assume that the number of sampled positions along each coordinate dimension is approximately proportional to the dimension length $S_i$, so we can represent the number of sampled positions along each dimension as $\mu S_i$, where $\mu$ is a scale factor. Because we require original sampling and our new sampling should have approximately the same number of sampled points in the same setting, we have:

$$\prod_{i=1}^{C} \mu S_i \approx N \tag{6}$$

By solving this equation, we can get

$$\mu = \sqrt[C]{\frac{N}{V}}, \quad V = \prod_{i=1}^{C} S_i \tag{7}$$

Though we limit the flexibility of the sampling process, the probability of choosing a particular point/region is the same for two sampling methods. To introduce more randomness to our sampling method, we also add random noise to each $\mu S_i$. Note that the sampled coordinate space can be continuous or discrete, but this split sampling method works in the same way.

As a result, this strategy ensures that the points selected can be fully decomposed along any dimension. All input points that are formed when the coordinates are fused are thus selected for training. This is depicted in Fig. 4.

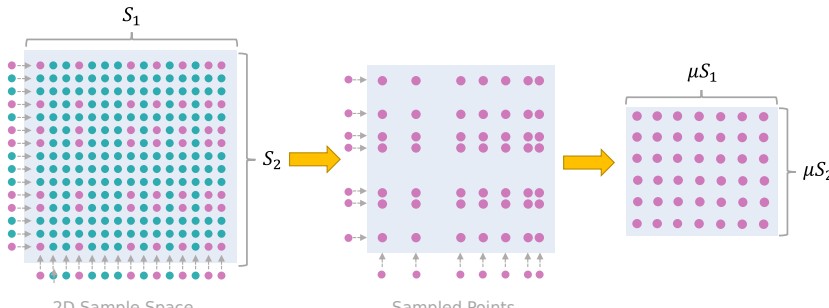

Figure 4: An illustration of our point sampling strategy in 2-D space. The data scale is $S_1, S_2$ on the first and second dimensions, our new sampling strategy ensures that the proportions of samples along each coordinate dimension are exactly the same and can be split along each dimension.

## B    Additional task details

In this section, we provide additional details and results regarding various fitting tasks evaluated in Section 4. All experiments on signal fitting tasks are implemented using PyTorch and use an NVIDIA RTX3090 GPU. We use the Adam optimizer (Kingma & Ba, 2014) during training.

### B.1    Image Fitting

This task involves training the model to fit 2D images. Given a 2D pixel coordinate, the model predicts the color of the pixel at that coordinate. One of the biggest advantages of using implicit

representations is that it can represent images with an arbitrary resolution, since it learns a continuous representation of the image, therefore it can represent an image at high resolution but with a size smaller than the image itself. We formulate this task as:

$$\Phi : \mathbb{R}^2 \mapsto \mathbb{R}^O \tag{8}$$

Where $O$ is the dimension of the pixel color value of the image, for example $O = 3$ if image is represented in RGB value, or $O = 1$ if it is a greyscale image.

In this experiment, we try to fit RGB images of size $512 \times 512$. We randomly select 12 center-cropped images from DIV2K dataset (Agustsson & Timofte, 2017) for this experiment, and train for 20,000 epochs.

In the image fitting tasks using PE, positional encoding maps input point $\mathbf{x} \in \mathbb{R}^K$ that is normalized to the range $[-1, 1]$ to $\mathbb{R}^{2Kd+K}$ using a set of sinusoidal functions:

$$\gamma(p) = (\sin(2^0 \pi p), \cos(2^0 \pi p), ..., \sin(2^{d-1} \pi p), \cos(2^{d-1} \pi p)) \tag{9}$$

Where $K$ is the dimension of each input point, $p$ is an individual coordinate in input point $\mathbf{x}$, and $d$ is the encoding frequency. We then concatenate this encoding with the original input $\mathbf{x} \in \mathbb{R}^K$, to get an input of dimension $2Kd + K$. We follow the settings and the encoding frequency used in the image fitting task introduced in (Tancik et al., 2020).

We separately sample 2D pixel coordinates on each axis. We sample 512 points from both axes for each training batch. Since the image size is $512 \times 512$, we can fit all sampled points into one batch on an RTX3090 GPU.

In Fig 5, we choose 4 images of various complexities from the DIV2K dataset to show the qualitative results for the image fitting task.

## B.2 Video Fitting

We also train our model to learn video representations. Compared to images, videos have an extra dimension which is the temporal dimension or the frame number. Therefore this fitting task has an input of 3D coordinates $(x, y, t)$ where $(x, y)$ is the pixel spatial location, and $(t)$ is the temporal location of that pixel, or the frame number it is at. Similar to the image regression task, we formulate this task as:

$$\Phi : \mathbb{R}^3 \mapsto \mathbb{R}^O \tag{10}$$

Where $O$ is the dimension of the pixel color value of the video at the given pixel location at the given frame. We also supervise the model using MSE loss between the predicted color and ground truth color. There are two splitting strategies. In the first strategy, we split all three dimensions of the input coordinates, for a set of coordinates in the format $(x, y, t)$, we split it into $(x)$, $(y)$ and $(t)$ and sample values in each separated channel. We denote this strategy as X+Y+T. In the second strategy, we only separate the pixel/spatial dimension with the frame/temporal dimension, for a set of coordinates in the format $(x, y, t)$, we split it into $(x, y)$ and $(t)$ and sample values in each separated channel. We denote this strategy as XY+T.

In this experiment, we use 5-layer coordMLPs with 1024 hidden units to fit 2 10-second video clips[4], and we train for 100,000 epochs.

While performing our sampling strategy for CoordX with XY+T splitting mentioned in Sec. 4.1, for each training batch, we first randomly permute frame values and choose the first $t$ values to be the frames to sample. We sample $3.8\%$ of the total points each batch, therefore for each frame, we sample $y = 0.038N/t$ points (the number is rounded to the nearest integer), where $N$ denotes the total number of points in the 3D space for the video. The coordinate values along the spatial (XY) dimension and the chosen frame values along the temporal (T) dimension for each sampled point are fed to CoordX. The sampling method for the X+Y+T strategy is very much similar, except each chosen point is further split along the XY dimension. Like the image fitting task, the model is supervised using an MSE loss of predicted color and ground truth color for each pixel.

We provide the qualitative results for the video fitting task in Fig 6 and 7, where we capture and show several frames of the two videos for comparison.

---

[4]A bike video from http://www.scikit-video.org/stable/datasets.html, and a cat video from https://www.pexels.com

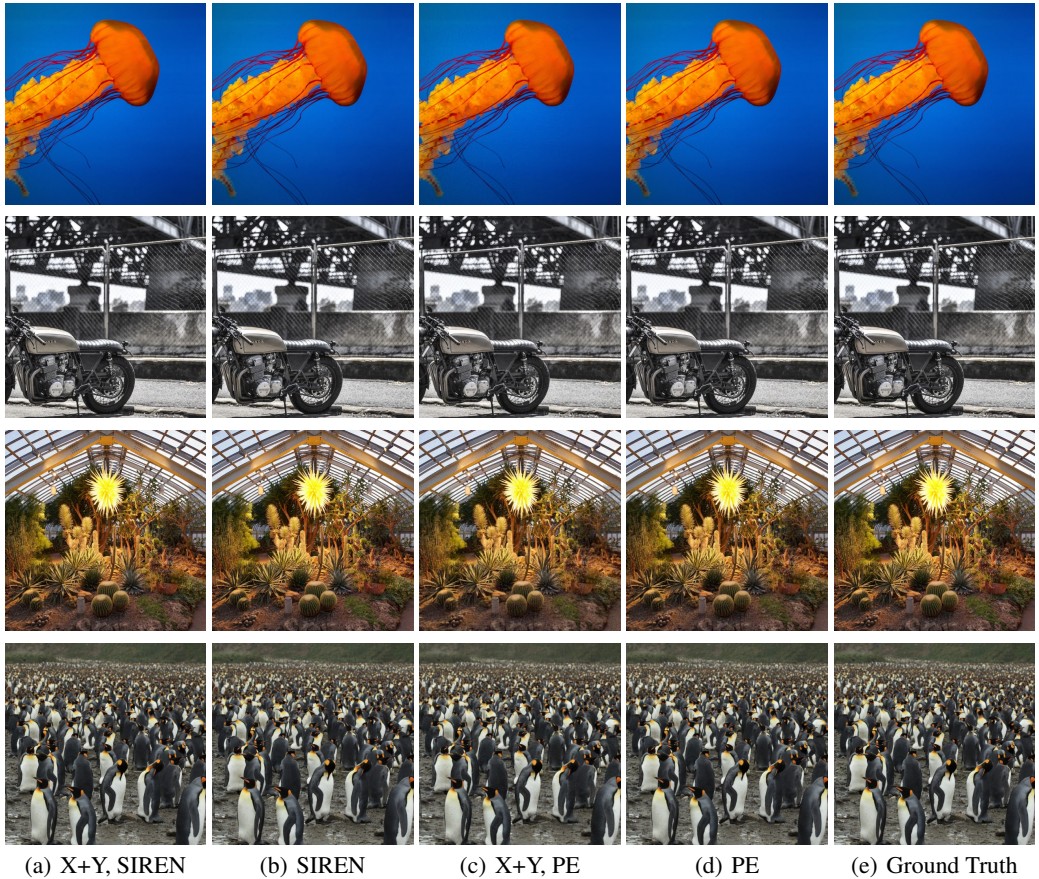

(a) X+Y, SIREN     (b) SIREN     (c) X+Y, PE     (d) PE     (e) Ground Truth

Figure 5: Qualitative results for the image fitting task. The 4 images shown are from the DIV2K dataset, center-cropped and down-sampled to a 512x512 resolution.

### B.3 3D MODEL FITTING

We represent 3D shapes using a 1D occupancy grid (Mescheder et al., 2019), inspired by similar works in (Tancik et al., 2020). An occupancy grid represents the shape as a 1D occupancy probability for a given 3D coordinate and indicate whether the query points are inside or outside the 3D shape:

$$\Phi : \mathbb{R}^3 \mapsto \mathbb{R}^1 \tag{11}$$

For this task, we split the input coordinates along all three dimensions. We sample values along each of the three coordinate channels of the input 3D coordinates. We use 5-layer coordMLPs with 256 hidden units to fit 4 3D objects (Bunny, Armadillo, Dragon and Buddha) from The Stanford 3D Scanning Repository[5], and train each for 10,000 epochs. We follow the settings in Tancik et al. (2020), and use cross-entropy loss for the model to learn the classification of occupancy labels.

We first recenter points loaded from mesh points to $[-1, 1]$ for SIREN test. For each batch, we randomly choose a number of points to sample per axis. We store those sampled points and their corresponding ground truth occupancy value in a file, to skip the time-consuming data sampling step, which acts as a major performance bottleneck in the training process, in order to present a more accurate results of our split acceleration.

---

[5]http://graphics.stanford.edu/data/3Dscanrep/

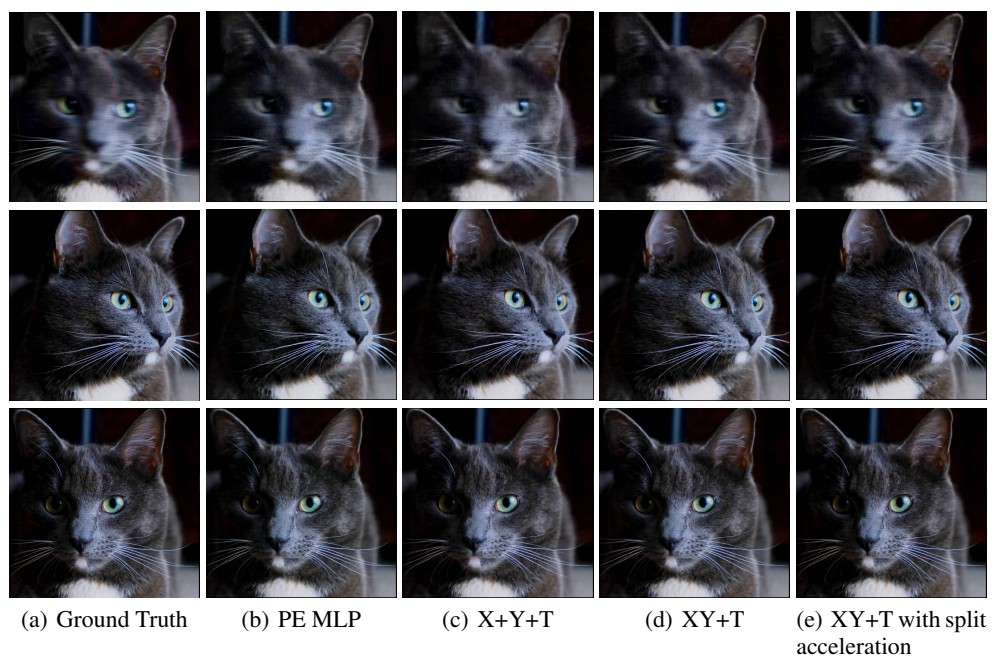

(a) Ground Truth     (b) PE MLP     (c) X+Y+T     (d) XY+T     (e) XY+T with split acceleration

Figure 6: Qualitative results for the video fitting task (the cat video).

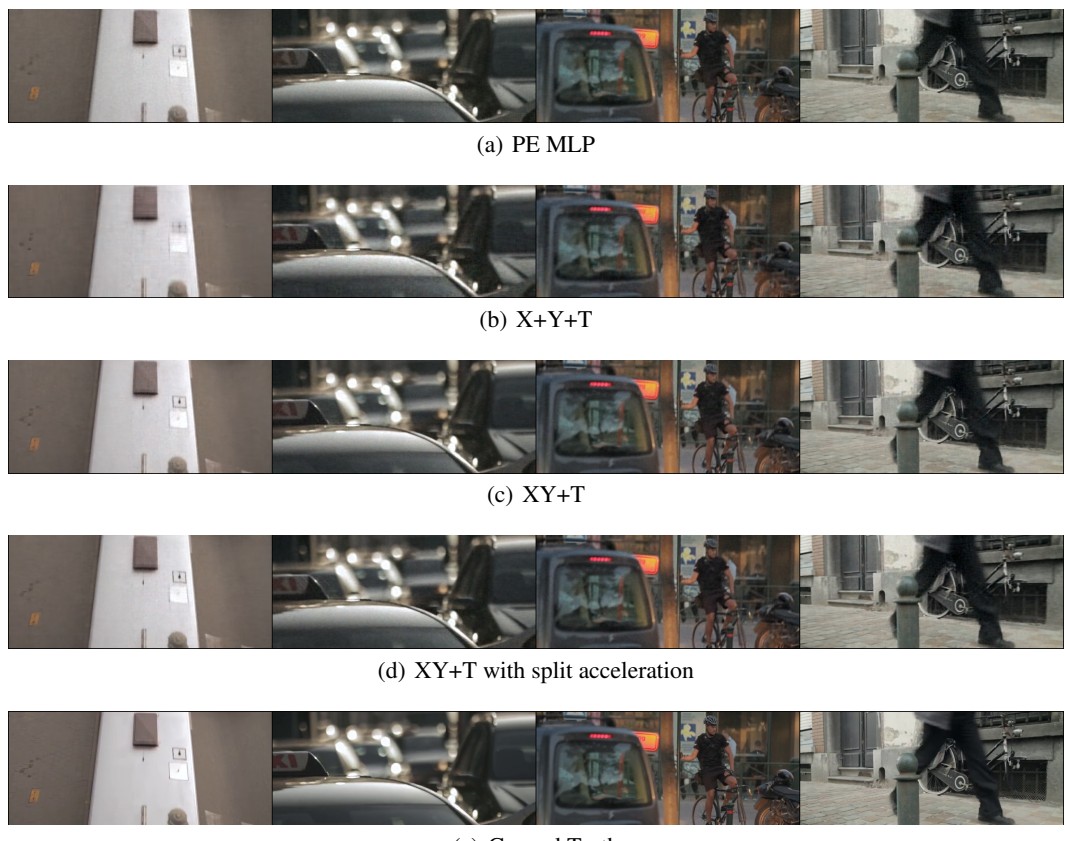

Figure 7: Qualitative results for the video fitting task (the bike video).

### B.4 FEATURE FUSION

In 3.1 while we choose multiplicative modulation (outer product) as our feature fusion strategy, here we present a comparison between our approach against additive modulation (summation) and concatenation.

Assuming fusion occurs after the $D_s$-th layer and we split the input into $C$ branches, additive modulation and concatenation strategy first reshape each feature $\mathbf{H}_{D_s}^{(i)} \in \mathbb{R}^{B_i \times M}$ from branch $i$ to become $\mathbf{H}'_{D_s(i)} \in \mathbb{R}^{B_1 \times \cdots \times B_C \times M}$ by expanding its dimension and repeating the feature along each new dimension with corresponding number of times. The fused feature for additive modulation strategy can then be formulated as follow:

$$\mathbf{H}_{D_s}^A = \sum_{i=1}^{C} \mathbf{H}'_{D_s}^{(i)} \tag{12}$$

Where $\mathbf{H}_{D_s}^A \in \mathbb{R}^{B_1 \times \cdots \times B_C \times M}$ is the fused feature generated by additive modulation.

The fused feature for concatenation strategy can be formulated as follow:

$$\mathbf{H}_{D_s}^{Cat} = \mathbf{H}'_{D_s}^{(1)} \oplus \cdots \oplus \mathbf{H}'_{D_s}^{(C)} \tag{13}$$

Where $\oplus$ denotes the concatenation between feature tensors along the last dimension, and $\mathbf{H}_{D_s}^{Cat} \in \mathbb{R}^{B_1 \times \cdots \times B_C \times (CM)}$ is the fused feature generated by concatenation.

The multiplicative and additive modulation doesn't require any modification to the FC layers after fusion, we simply retain the FC layers from the baseline model. Concatenation strategy requires modification to the first layer after fusion. We denote that layer as the $k$-th layer which originally has the formulation $\{\mathbf{H}_k = \phi(\mathbf{W}_k \mathbf{X})\}$ where $\mathbf{X} \in \mathbb{R}^{B_1 \times \cdots \times B_C \times M}$, $\mathbf{W}_k \in \mathbb{R}^{M \times M_{out}}$, and $M_{out}$ is equal to the number of hidden features in the subsequent layer, or the fitted signal dimension if $k$-th layer is the last layer in the network. Since input feature now has last dimension size $CM$ instead of $M$, the $k$-th layer is modified as $\{\mathbf{H}'_k = \phi(\mathbf{W}'_k \mathbf{X}')\}$ where $\mathbf{X}' \in \mathbb{R}^{B_1 \times \cdots \times B_C \times CM}$, $\mathbf{W}'_k \in \mathbb{R}^{CM \times M_{out}}$. This modification also increases the number of parameters in the $k$-th layer.

We follow the same experimental settings in 4.1 and perform image fitting tasks training for 10,000 epochs using the additive modulation, multiplicative modulation, and concatenation fusion strategies. Results are shown in Table 6. We observe that multiplicative modulation performs the best out of the three strategies. Additive modulation and multiplicative modulation use the same number of parameters, while concatenation requires more parameters than multiplicative modulation. This results in ∼8% increase in training time compared to multiplicative modulation.

| Method | Prod-S | Sum-S | Concat-S | Prod-P | Sum-P | Concat-P |
|--------|--------|-------|----------|--------|-------|----------|
| PSNR | 36.32 | 33.03 | 35.10 | 31.41 | 26.51 | 28.18 |

Table 6: Quality comparison between different fusion strategies. Prod, sum and concat denote multiplicative modulation, additive modulation and concatenation. S and P refer to SIREN and PE respectively.

## C VOLUME RENDERING WITH COORDX

Volume rendering is another important application of coordinate-based MLPs in 3D graphics visualization. Unlike the previously discussed tasks which show good input data alignment (i.e., the input coordinates can be easily decomposed along each dimension), the volume rendering application does not have this property in its sampled coordinates, e.g., Fig. 8. To enable CoordX's use in volume rendering tasks, we use an interpolation method that transforms the point sampling space into axis-aligned grids. The details of our solution will be discussed in the following sections.

### C.1 VOLUME RENDERING

NeRF (Mildenhall et al., 2020) performs ray rendering during training and inference. It essentially renders a 2D image based on the camera ray from a given camera location. This method is derived

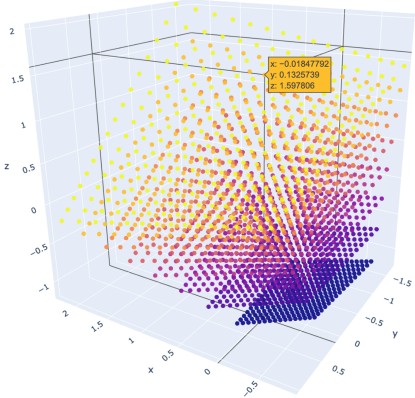

Figure 8: Illustration of sampled coordinate points for volume rendering.

from the classic volume rendering technique (Kajiya & Von Herzen, 1984) to render a 2D image given the camera location. The expected color $C(r)$ where $r(t) = o + td$ is the function representing ray from camera, with near and far bound $t_n, t_f$ can be described as:

$$C(r) = \int_{t_n}^{t_f} T(t)\sigma(r(t))c(r(t), d)dt \quad \text{where} \quad T(t) = \exp(-\int_{t_n}^{t} \sigma(r(s))ds) \qquad (14)$$

To estimate this continuous integral shown in the Eqn. 14, NeRF uses a volume sampling technique, where 3D points are densely sampled along each camera ray, the color and volume density at each point is accumulated and used to predict the final color of the camera ray in order to render the final image:

$$C(r) \approx \sum_{i=1}^{n} T(i)(1 - \exp(-\sigma_i\delta_i))c_i \quad \text{where} \quad T(i) = \exp(-\sum_{j=1}^{i-1} \sigma_j\delta_j) \qquad (15)$$

Here $\delta_i$ is the distance between adjacent point samples $t_i, t_{i-1}$, $c_i$ and $\sigma_i$ is the color and volume density of the sampled point.

In real applications, hierarchical sampling is used for higher-quality rendering. A coarse sampling is first performed so the rough depth information of the object can be obtained. Next, we do fine-grained sampling around the object surface based on estimated depth information.

## C.2 RAY SAMPLING WITH A SPLIT MLP ARCHITECTURE

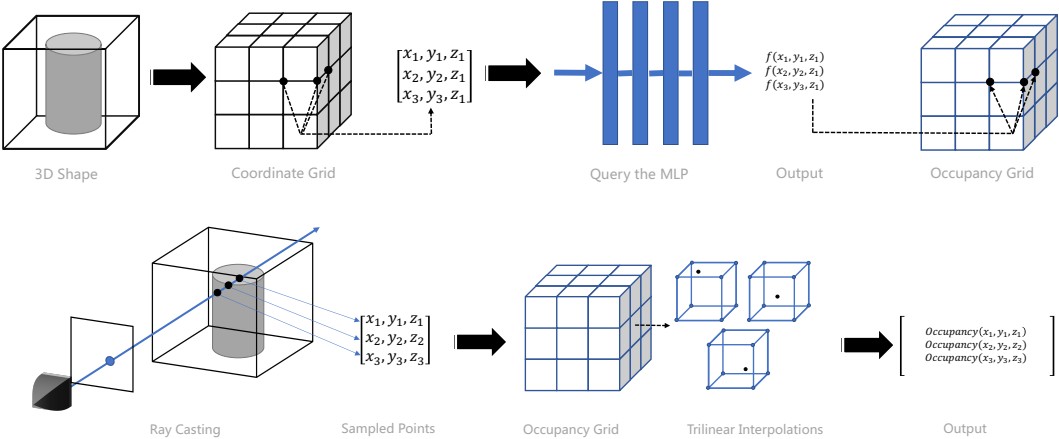

Figure 9: Illustration of our modified volumetric rendering.

As explained in C.1, volume rendering includes casting a ray from the camera to each pixel of the rendered image, and sampled points along that ray. After that, each sampled coordinate point

is used as the input to the coordMLP, and it predicts a signal for each point. These predictions are then accumulated with other points on the same ray using rendering functions to generate the final color of the ray at that pixel. The difficulty of incorporating our CoordX model with volume rendering is that the sampled points are not axis-aligned and cannot be decomposed along coordinate dimensions. Since the camera ray is usually not aligned with any of the $x, y, z$ axis in 3D space, sampled points along the ray do not share coordinate values (Fig. 8). To address this problem, we use an interpolation-based rendering method that outputs predicted signal values of queried points by interpolating on a pre-computed grid. The interpolation-based acceleration can also be seen in other relevant work such as FastNeRF (Garbin et al., 2021) and PlenOctrees (Yu et al., 2021).

As illustrated in Fig. 9, we first pre-compute a 3D occupancy grid for the 3D object with pre-defined resolution (e.g., 256). Each voxel inside the grid occupies a small cubic space. After pre-computation, if we are to render an image from an arbitrary camera angle, we can perform interpolation-based volumetric rendering, where the occupancy of the points we sampled along each ray is trilinearly interpolated from the 8 endpoints of the voxel cube which this point is in. We then get accumulated color/depth for each ray using the original volumetric rendering methodology.

Based on this interpolation raycasting method, we can replace the baseline CoordMLP with our CoordX model to accelerate the grid pre-computation step (the most time-consuming step in the whole processing pipeline). The points in the pre-computed 3D coordinate grid can be decomposed into 3 vectors which are used to query CoordX. We demonstrate a 2.8x speedup with CoordX in Fig. 3(c).

A similar approach can be used to accelerate training for NeRF models. However, using grid interpolation for training requires training with all the points in the grid, which can be much larger than the number of points required to train the baseline model. Thus our grid interpolation-based training may consume more memory and take longer training times. To address this problem, we decide to first compute the split features just before fusion for the corresponding grid. For each query point, bilinear interpolation is performed directly on these split features, the interpolated features will then be fed into the second part of our splitting model to predict final signal values. In this way, we strike a balance between grid reuse and computation efficiency.

### C.3 VISUALIZATION OF LEARNED 3D OCCUPANCY.

We apply our proposed rendering method to MLP models for 3D occupancy representations trained in 4.1. We choose the original volumetric rendering method as the baseline rendering method. We choose $512 \times 512$ as the target resolution of the rendered images. We uniformly sample 256 points along the ray for each level of two granularity levels in hierarchical rendering. The coordinate grid used in our grid interpolation rendering method has the resolution of 384 for the longest length of the object's 3D bounding box. Fig. 9 shows the qualitative comparison of the rendered images. Our grid interpolation-based rendering is referred to as grid in the figure. Table 7 shows the speed comparison of different rendering methods. We test the time required to render one image (including the time to pre-compute the grid for our method).

We make the following observations. First, grid interpolation rendering can achieve significant speedups compared to baseline volumetric rendering (2.70x and 14.83x for baseline and CoordX respectively). Second, with our grid interpolation rendering, CoordX achieves a 2.15x speedup over the baseline MLP. Third, the rendered images in Fig. 9 show our rendering method is able to render high-quality images for different MLP models.

| Model | baseline MLP | | CoordX | |
|---|---|---|---|---|
| Rendering Method | baseline | grid | baseline | grid |
| $T_{\text{rendering}}$ (sec) | 6.61 | 2.45 | 16.91 | 1.14 |

Table 7: Rendering time comparison for different volumetric rendering method.

### C.4 NOVEL VIEW SYNTHESIS

We train a simplified 8-layer NeRF model for three blender scenes (Lego, ship and drums). We use the spherical harmonics (SH) function proposed in Yu et al. (2021) to decompose the point position

and view direction so that the input to the NeRF MLP is just the positional coordinates $(x, y, z)$. We refer to this model as our baseline NeRF model. Our CoordX NeRF model is based on this baseline model. We split the positional coordinates $(x, y, z)$ in 3 branches and 2 FC layers are kept after the fusion operation ($D_s = 6, D_f = 2$). The resolution of the feature grid used for bilinear interpolation is set to 384.

To train these NeRF models, we use downsampled ground truth images ($400 \times 400$) for supervised learning. 128 points are sampled along randomly selected rays. For simplicity, we do not use hierarchical sampling for NeRF models. The models are trained for 200,000 epochs. To visualize the scene learned by NeRF models, we evaluated both baseline volumetric rendering and grid interpolation rendering (similar to 3D shape rendering). The target resolution of the rendered images is $400 \times 400$. The coordinate grid used has a resolution of 384 for the longest length of the object's 3D bounding box. We sample 256 points along the ray (without hierarchical sampling). The PSNR is evaluated based on these rendered images. Table 8 compares the baseline NeRF and CoordX NeRF from different aspects, and Fig. 11 and Fig. 12 shows the rendered images for qualitative comparison.

| Metrics | | PSNR | | | $T_{\text{rendering}}$ (sec) | $T_{\text{training}}$ (hr) |
|---------|---------|------|------|-------|------------------------------|-----------------------------|
| | | Lego | Ship | Drums | | |
| baseline-NeRF | baseline | 29.65 | 28.28 | 24.77 | 3.09 | 4.73 |
| | grid | 29.23 | 27.93 | 24.61 | 1.97 | |
| CoordX-NeRF | baseline | 29.29 | 28.05 | 24.84 | 7.82 | 2.61 |
| | grid | 29.29 | 27.75 | 24.76 | 0.70 | |

Table 8: Quantitative comparison between baseline NeRF and our CoordX NeRF. The speeds in the table are tested on the Lego scene.

We make the following observations. First, our proposed interpolation-based training can help CoordX achieve a 1.81x training speedup over baseline NeRF. Second, our CoordX NeRF model achieves similar PSRN scores (less than 0.5 dB drop) compared to the baseline NeRF. Third, our grid interpolation rendering method can achieve the same level of rendering accuracy (less than 0.5 dB drop) compared to the original rendering method for both baseline and CoordX models.

In summary, we demonstrate that our proposed grid interpolation method is able to render images with comparable quality with significant rendering speedups. This method also enables our CoordX model to achieve training and inference acceleration over the baseline model.

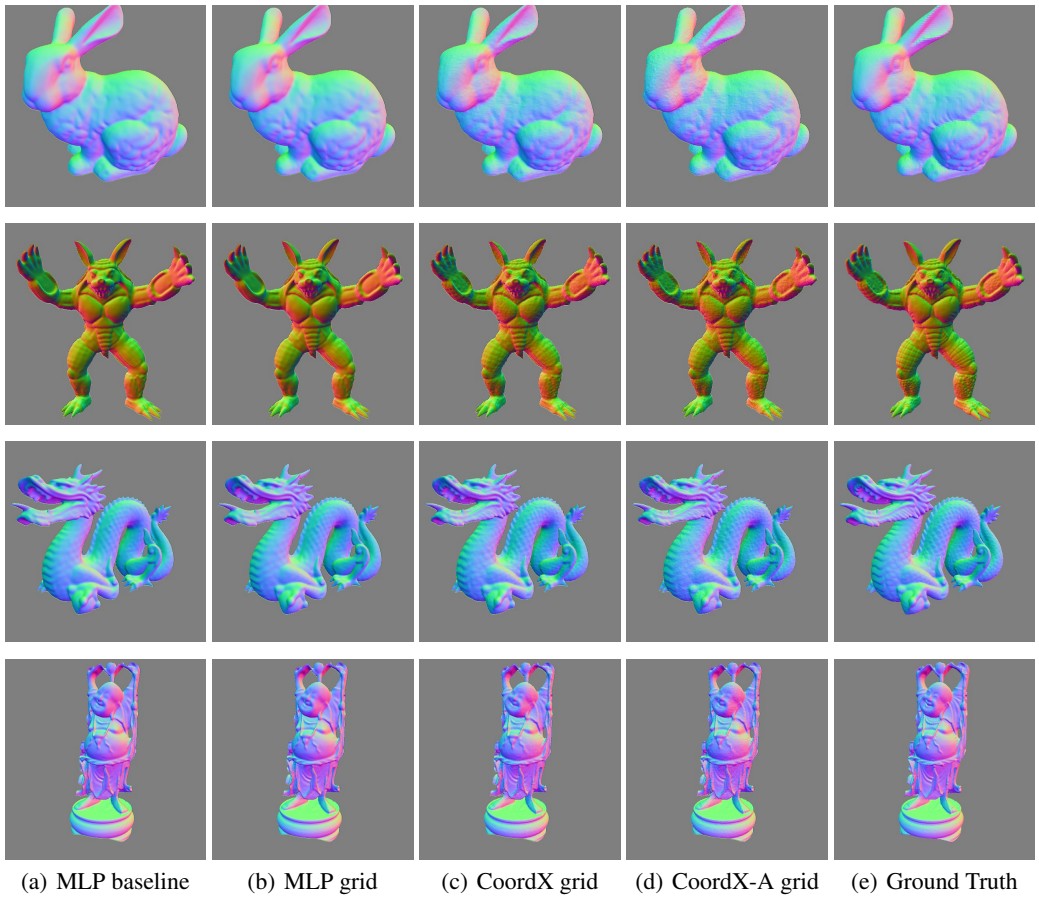

| (a) MLP baseline | (b) MLP grid | (c) CoordX grid | (d) CoordX-A grid | (e) Ground Truth |

Figure 10: Images result rendered using different ray rendering methods.

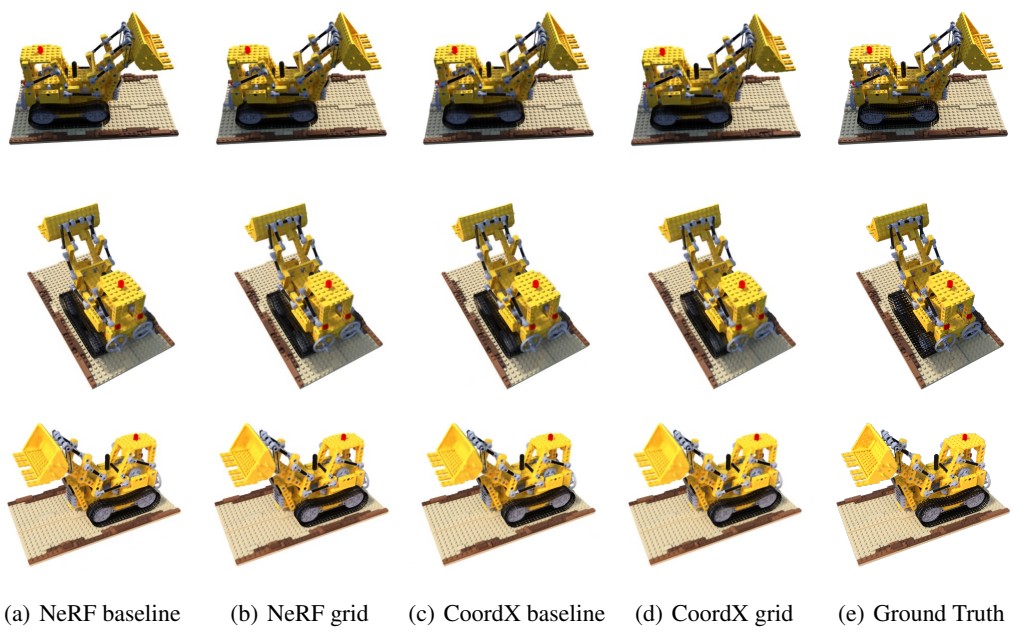

| (a) NeRF baseline | (b) NeRF grid | (c) CoordX baseline | (d) CoordX grid | (e) Ground Truth |

Figure 11: Images rendered using NeRF on Lego dataset.

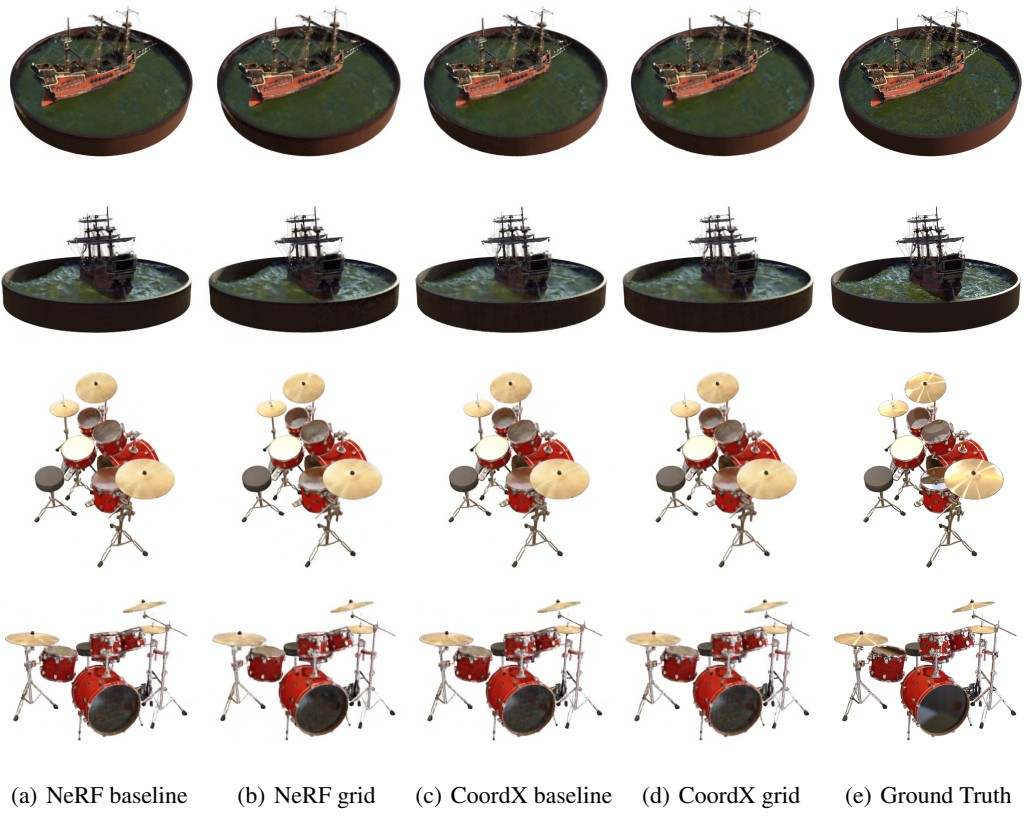

(a) NeRF baseline     (b) NeRF grid     (c) CoordX baseline     (d) CoordX grid     (e) Ground Truth

Figure 12: Images rendered using NeRF on Ship and Drums dataset.

