# OpenReview forum: "CoordX: Accelerating Implicit Neural Representation with a Split MLP Architecture"
_ICLR.cc/2022/Conference — ICLR 2022 Poster_

### Official Review · Reviewer_z7hu · 2021-10-29

**Correctness:** 3
**Technical Novelty And Significance:** 3
**Empirical Novelty And Significance:** 3
**Recommendation:** 6
**Confidence:** 4

**Main Review:**

While I think the motivation of this paper is interesting and I appreciate the many experiments the authors have run, I still believe there are quite a few issues with the paper which need to be addressed. The main strengths and weaknesses (in my eyes) are described below.

**Strengths**:
- The motivation of the paper is good and improving the speed of implicit reps is an important topic. The observation that building methods that leverage the locality between coordinate points could help implicit reps (as opposed to regular implicit reps that consider points as independent elements of a batch) is interesting
- The authors perform experiments on a wide variety of data modalities (images, videos, 3D shapes, NeRF scenes) and make some interesting ablations

**Weaknesses**:
- In general, the experimental results are not very convincing. For example, on the image experiments, the training time can be halved but at the cost of a 3dB drop in PSNR. The authors refer to this as “a slight drop in quality” but 3dB is a very significant drop (PSNR is measured on a log scale). For videos and the superresolution experiments the reconstruction results are on par with the baselines but are faster, which is good. However, comparisons to e.g. [1] or other methods aiming to accelerate implicit reps are missing. Methods such as these would very likely both be faster and have better reconstruction accuracy than the proposed method. Adding ablations against more than just vanilla implicit reps models would definitely help strengthen the paper.
- One of the main drawbacks of the proposed method is that the decomposition operation requires a regular grid. A huge part of the motivation for using implicit representations is exactly to be able to model data that does not lie on a regular grid, e.g NeRF scenes, meshes and so on. So this does seem to be a significant drawback. While the authors briefly discuss this and propose a fix for NeRF scenes, this fix still relies on the existence of some underlying grid (256^3) which then limits the resolution of the signal.
- The related work section disregards some very closely related work in accelerating training + inference of implicit representations [1]. While the authors cite this paper among others, they then mention that competing methods either are only applicable to a single task (not the case for [1]), have high memory overheads (not the case with [1], in fact memory is reduced) and they are not effective for dense/rich signals like images (not the case for [1], which can model gigapixel images). While [1] can arguably be considered concurrent work, I believe there should be a more fair discussion of this.
- The paper writing is not very clear. The authors introduce a lot of notation which can make the paper heavy to read. In addition, the authors seem to use very non-standard notation. For example, in equation (1) and (2) it appears like the authors use ; to denote matrix multiplication which I’ve never seen before. In equation (3) the outer product is denoted with a typical product symbol, it would be much clearer to just write this as an outer product.
- The paper seems rushed and there are many typos. For example, it looks like the first sentence of the paper is misplaced.
- Comparisons on e.g. NeRF scenes do not seem fair as there is a lot of work accelerating the rendering of NeRF scenes which considerably outperforms the proposed method (including the papers [2, 3] that the authors already cite). It would be nice to add comparisons to these too.

The authors also mention in the conclusion that decomposing signals into smaller signals that can be used to reconstruct the original signal could potentially be applied to data compression. This was one of the first thoughts I had when initially reading the paper and I think it would be interesting to expand more on this, e.g. in the context of [4].

[1] ACORN: Adaptive Coordinate Networks for Neural Scene Representation, Martel et al.

[2] FastNeRF: High-Fidelity Neural Rendering at 200FPS, Garbin et al.

[3] PlenOctrees for Real-time Rendering of Neural Radiance Fields, Yu et al.

[4] COIN: COmpression with Implicit Neural representations, Dupont et al.


**Summary Of The Paper:**

This paper proposes a new architecture for implicit neural representations, called CoordX, which splits each dimension of the input signal into separate branches (e.g. the x and y coordinates of pixel locations in an image) and processes each of these separately before fusing them. The authors achieve this by projecting each of these branches into a hidden feature and then using shared fully connected layers to process these. Each branch is then fused by an outer product which then reconstructs the entire input grid (e.g. for an image of size H x W, 2 branches take in H locations and W locations respectively which are then combined into H x W features by the fusion operation). The fused layers are processed by a few more MLP layers to output the predicted features. In addition, the authors propose a method for effectively subsampling the grid during training as well as different splitting strategies for the branches.

The authors perform experiments on various data modalities, including images, videos, 3D shapes and NerF scenes.

The main contributions of the paper in my eyes are then:
- Introducing a new architecture for implicit neural representations that in certain cases can improve training/inference speed without incurring a decrease in reconstruction
- Experiments on various data modalities showing the strengths/weaknesses of the method


**Summary Of The Review:**

This paper introduces a new architecture for implicit neural representations. While there are some interesting ideas in the paper, overall I still believe more work needs to be done and better comparisons to baselines need to be made before this paper is ready for publication.

--- Post rebuttal update ---

I appreciate the large efforts the authors have put in to address the issues put forward in this review and by the other reviewers. With these changes I believe the paper is stronger and have updated my score accordingly. In particular I appreciate the experiments showing that CoordX can also help approaches such as ACORN (even though these results still seem preliminary, particularly for Tokyo). Including these results in the updated paper will definitely help give a clearer picture of the contributions. The extended discussion and experiments around NeRF scenes are also helpful and appreciated.

While I still believe there are some drawbacks with the proposed approach (in particular that using the method for non grid-based signals requires slightly hacky workarounds and that the gains by the proposed method are quite small compared to other methods aiming to accelerate implict reps e.g. ACORN), I am happy for this paper to be accepted in its current updated state.

---

> ### Author Response · Authors · 2021-11-16
> **Response to Reviewer z7hu (1/2)**
>
> Thank you for your detailed comments and appreciation of our experiments. We respond to your concerns and questions below.
>
> ### Weakness 1
> * We have fixed the verbiage “a slight drop in quality” in Section 4.1. The 3db PNSR can be alleviated by essentially adding more parameters to the CoordX model at the cost of a higher execution time compared to the baseline CoordX model. However, even when adding more parameters to CoordX, we achieve significantly faster training and inference times compared to the baseline MLP model without losing any accuracy. This is described and evaluated in Section 4.4.
>
> * The second point mentioned by the reviewer is the lack of comparison against other concurrent works such as ACORN [1]. Our method is essentially orthogonal to other concurrent optimization methods that leverage explicit structures (e.g., quadtree and octree in [1]) to improve training/inference speed -- CoordX can be integrated into the coordinate encoder in ACORN to generate additional speedups. We use the vanilla coordinate-based MLP (with SIREN or PE) as the baseline model to integrate CoordX in our evaluation. Thus, the concurrent methods with explicit structures can outperform our models in terms of speed and accuracy. However, our method can be combined with these methods to provide additional speedups, similar to what we did in LIIF experiments in Sec. 4.2. We were unable to provide experimental evaluations to show how the CoordX method can accelerate concurrent optimized methods (specifically [1] and [2]) for further speedup since the official code release of [1] or [2] was unavailable in time for evaluation before submission (the code in [2] is yet to be released). However, our very recent initial evaluations of integrating CoordX into ACORN [1] demonstrate that we were able to accelerate ACORN’s coordinate encoder when fitting large-scale images (8K Pluto and 1GB Tokyo). The following table shows PSNR values and the time spent on the coordinate encoder during inference.
>
>   |                | PSNR-Pluto | $T_{\text{encoder}}$-Pluto | PSNR-Tokyo | $T_{\text{encoder}}$-Tokyo |
>   | -------------- | ---------- | -------------------------- | ---------- | -------------------------- |
>   | baseline-ACORN | 40.82      | 61 ms                      | 22.81      | 233 ms                     |
>   | CoordX-ACORN   | 40.77      | 42 ms                      | 22.78      | 173 ms                     |
>
>   These results are from our optimized ACORN code, we have removed redundant encoder queries from their original code, otherwise, the baseline ACORN will take a longer time on encoder inference. The 8K Pluto model is trained for 50K iterations, The Tokyo model is trained for just 5,000 iterations (this task is very time-consuming and memory-consuming). We can see from our current results that our CoordX method can provide additional speedups to the ACORN’s coordinate encoder (1.45x for Pluto and 1.34x for Tokyo), with less than 0.1dB PSNR drop. We will provide more details of this evaluation in our code release.
>
> ### Weakness 2
> * We agree that relying on a regular grid of input is a limitation of our method. Due to this limitation, we cannot directly use our method to accelerate volume rendering tasks such as NeRF by decomposing the input xyz coordinates. Appendix C just shows an initial workaround for volume rendering tasks that generate speedups over the baseline NeRF model. While other concurrent NeRF-specific acceleration works [3,4] achieve higher speedups, these techniques are orthogonal to CoordX and CoordX can be used to achieve even further speedups. In addition to the CoordX technique for NeRF proposed in Appendix C, there are other solutions that can potentially enable significantly higher speedup for NeRF-based volume rendering using CoordX. For example, we can use different coordinate systems to represent points on the camera rays (e.g., use (o, d, t) instead of p(x,y,z), as p = o+td ) or we can directly parameterize the camera rays (e.g., [3]). We leave these explorations for future work.
> * It should be noted that for dynamic NeRF models in which time index t or global feature vectors are added as additional input dimensions of NeRF’s MLP [4,5], CoordX can generate higher speedups than for the static models. Our input coordinate decomposition method can significantly reduce the amount of computation for these extra input dimensions, similar to our XY+T video experiment (Sec. 4.1).
>
> ### Weakness 3
> * We have added a more detailed discussion of ACORN [1] to Section 2.
>
> ### Weakness 4 & 5
> * We thank the reviewer for pointing out these issues. We have fixed the issues with notation and typos in our revised submission.

---

> > ### Author Response · Authors · 2021-11-16
> > **Response to Reviewer z7hu (2/2)**
> >
> > ### Weakness 6
> > * We present a simple solution to apply CoordX to accelerate the baseline NeRF model in Appendix C which provides some reduction in training and inference time. When used for NeRF, CoordX however does not provide the same speedups as the concurrently-proposed NeRF-specific acceleration techniques [6, 7] which pre-compute the prediction results with densely sampled points before starting the rendering (thus there is no MLP inference during the rendering process). CoordX however is orthogonal to these techniques and our method can be integrated with these acceleration methods to reduce the time spent on the MLP queries of densely sampled points in the pre-compute stage.
> >
> > ### Other Suggestions
> > * Thanks for the suggestions. [4] propose to use a neural network for image compression. In their official code release, they also use SIREN [9] as the implicit representation to overfit images and mentioned the slow compression speed as their main limitation. Therefore as shown in sections 4.1 & 4.4, we can achieve significant speedup using CoordX. We agree that data compression would be an interesting direction for further research, we will explore more on the topic of high compression + faster reconstruction in our future research. We added a citation to [8] under Section 5.
> >
> > [1] ACORN: Adaptive Coordinate Networks for Neural Scene Representation, Martel et al.
> >
> > [2] Modulated Periodic Activations for Generalizable Local Functional Representations, Mehta et al.
> >
> > [3] Light Field Networks: Neural Scene Representations with Single-Evaluation Rendering, Sitzmann et al.
> >
> > [4] Neural 3D Video Synthesis, Li et al.
> >
> > [5] AD-NeRF: Audio Driven Neural Radiance Fields for Talking Head Synthesis, Guo et al.
> >
> > [6] FastNeRF: High-Fidelity Neural Rendering at 200FPS, Garbin et al.
> >
> > [7] PlenOctrees for Real-time Rendering of Neural Radiance Fields, Yu et al.
> >
> > [8] COIN: COmpression with Implicit Neural representations, Dupont et al.
> >
> > [9] Implicit neural representations with periodic activation functions

---

> > > ### Comment · Reviewer_z7hu · 2021-11-18
> > > **Thank you for your reply**
> > >
> > > Thank you very much for your detailed reply! I have updated my score accordingly and have also written a post rebuttal update to the review (see Summary Of The Review section).

---

### Official Review · Reviewer_xWSd · 2021-11-01

**Correctness:** 4
**Technical Novelty And Significance:** 3
**Empirical Novelty And Significance:** 2
**Recommendation:** 8
**Confidence:** 4

**Main Review:**

The strengths of the paper are as follows:
1. The quantitative results are communicated clearly, and the benefits of the method are well justified and ablated.
    - The benefits in speed and lack of degradation in quality are shown across three signal fitting tasks (image, video, shape representation). It is also demonstrated that this idea can work for local implicit methods (using latent codes) and still provide a speed-up with limited degradation in quality.
    - The degree of splitting versus number of layers comparison is insightful and shows that the method proposed is in fact more powerful than simply using smaller MLPs (i.e. the individual processing FC blocks for each coordinate are highly useful).
2. The contribution of the paper is quite clear. I think that the simple idea of splitting coordinate processing, ablating it well and showing that it doesn’t degrade quality but speeds up the training and evaluation is quite clear. This makes it much more likely to be more widely adopted in practice, as faster training of coordinate-based representations of signals is a very important problem with lots of recent work.

In my opinion, the weaknesses of the paper are as follows:
1. The exposition of the significance of the results, and the necessary contributions to obtain them.
    - While fitting a single image, video, or 3D signal with a coordinate-based network is intellectually interesting as a toy problem, the main use of coordinate-based networks is in neural rendering. This means that it is essential for CoordX to be able to be applied to the volume rendering of NeRF-like methods, or the ray-tracing applied in neural surface based methods. However, to find any mention of volumetric methods using CoordX, I had to get to section C in the appendix. I think that this should absolutely be included and emphasized in the main paper, and ideally more experiments should be dedicated to this instead of only overfitting on the Blender lego scene. Additionally, this section should include more ablation on the properties of the modified volumetric rendering proposed. Without this, the magnitude of the contribution is severely limited, as simply being able to overfit a few types of signals faster isn’t that impactful to many applications.
    - Similarly, section A of the appendix should be mentioned in a bit more detail in the main paper. The input point sampling is crucial for actually obtaining the speedup benefits of the CoordX architecture (just doing the same minibatch training as standard coordinate-based networks would not yield any speedup), and thus the contribution of CoordX should be both the modified architecture and the sampling method.
2. Following weakness 1, first bullet point, I think that while the evaluation on the lego NeRF model is very crucial to include in the paper, I think that this should be studied more and more results should be demonstrated on more varied datasets. This would make the paper significantly stronger, because then CoordX architecture could be applied in many neural volume rendering systems, which have had a significant amount of work in recent years.

Minor comments:
- I think that the introduction could be reordered to set a better context for the contribution so as to not make it immediately confusing. For example, starting the first sentence with what CoordX does doesn’t make sense since I haven’t read about CoordX yet. Starting by introducing coordinate-based MLPs and their applications makes it more clear in what area the contribution of the paper will come from.
- The notation is a bit confusing in equation (2). Shoudn’t $W_j$ also have an $i$ index, since this is an independent weight matrix for each branch?
- It would be nice to have an ablation on the feature fusion method, as concatenation versus multiplicative modulation versus additive modulation has been a topic in generalizing over coordinate-based networks. This could certainly be interesting and enlightening on whether or not this is a crucial part of the CoordX architecture.


**Summary Of The Paper:**

The paper proposes a novel coordinate-based network architecture which proposes to process each of the input coordinates independently in the first layer instead of together in a fully connected layer. This input style results in a speed-up in terms of evaluation of the network, and thus faster training and inference in tasks where coordinate-based MLPs are used, without incurring a significant degradation in terms of the quality of the signal fit. These benefits are demonstrated for the tasks of image representation, video representation, and 3D shape representation.

**Summary Of The Review:**

Overall, the paper proposes an architecture modification which improves the evaluation time of coordinate-based MLPs without significantly hurting the quality of the signal fit. This contribution is demonstrated ablated clearly for signal memorization, and I believe that the proposed architecture has value for direct signal fitting tasks in coordinate-based networks. However, the effectiveness of the method is not immediately clear for applications in neural rendering, where coordinate-based networks are mainly used. A novel formulation of volume rendering which is compatible with the architecture change is proposed, and evaluated on one scene, which provides promising results that this can speed up the training of NeRF-like models in neural rendering, but this section of the appendix could be expanded upon significantly to create a significantly stronger paper. Based on the current state of the paper, I believe that the architecture shows promise and is ablated well for signal memorization, but may not be as significant as it could otherwise be. Thus, I believe it is marginally above the acceptance threshold.

**Post Rebuttal Update**

The authors have done a good job of addressing my concerns in my original review. I think that while the paper does propose a small algorithmic change and relies on more complex sampling methods to work for general coordinate-based network applications (as noted by other reviewers), the results and contributions are ablated well and the simplicity of the idea could likely have a larger impact upon practical usage of coordinate-based networks. Thus, I have updated my score, and think that the paper in its current state should be accepted.

---

> ### Author Response · Authors · 2021-11-16
> **Response to Reviewer xWSd**
>
> Thank you for your comments and suggestions about our paper. We respond to your concerns and questions below.
>
> ### Weakness 1
> * We thank the reviewer for the feedback. We have incorporated a discussion of using CoordX for NeRF and the evaluation results in the main text (Section 4).
> * Thanks for your suggestions on our sampling method. We have added more discussion about point sampling to the main text in our revised submission (Section 3.2).
>
> ### Weakness 2
> * We tested two more 3D scenes (ship and drums) last week and will update the draft with these collected results.
>
> ### Minor comments
> * Thanks for pointing out some writing issues, we will fix them in the revised submission.
> * Equation (2) shows the formulation of the shared layers after the first split layers. After the first layer, each feature tensor passes through the same shared layers, we use the shared weight matrices for all features across all branches, therefore $W_j$ does not have $(i)$ index, because it’s shared between all branches, not independent between each branch, and the $j$ here represents that $W_j$ is $j$-th layer’s weights.
> * Thank you for your suggestion on the ablation study of fusion method comparison. We choose the outer product in order to give more explanation from the perspective of matrix decomposition, but we also agree that such an ablation would be helpful. Therefore, we did such an experiment in the past few days, the results were added to the revised submission under Appendix B.4.

---

> > ### Comment · Reviewer_xWSd · 2021-11-18
> > **rebuttal response**
> >
> > I appreciate the response and updates to the paper. I have also updated my score and written a small post-rebuttal summary.

---

### Official Review · Reviewer_KUjc · 2021-11-02

**Correctness:** 4
**Technical Novelty And Significance:** 2
**Empirical Novelty And Significance:** 2
**Recommendation:** 6
**Confidence:** 3

**Main Review:**

I have no major issue with the paper. The idea is clearly presented with nice qualitative and quantitate support. The matrix decomposition discussion in Section 3.3 is pretty interesting to read. The algorithmic novelty, however, is rather limited. It's incremental, but delivers good results.

**Summary Of The Paper:**

The paper proposes an interesting tweak to the network architecture to accelerate CoortMLP. The idea is to split the input coordinates along the dimensions and then share weights before fusion.  The authors analyze the theoretical upper bound (as far as the MAC ops are concerned) and show about 2X speedup on actual machines.

**Summary Of The Review:**

Novelty is limited. Practical engineering efforts with wide applicability and good results.

---

> ### Author Response · Authors · 2021-11-16
> **Response to Reviewer KUjc**
>
> Thank you for your feedback, we are glad that you found our work interesting and that it delivers good results. As for the concern about novelty, we believe that our work provides a new way to accelerate coordinate-based MLPs which does not appear in prior works. No prior work has investigated the decomposition of input coordinates as a means to reduce computation in coordinate-based MLPs. Compared to other works on the acceleration of implicit neural representations, our idea is also more general and can be used orthogonally to provide further speedup in addition to the speed up achieved by those works. In addition, our method provides a new idea on multidimensional signal/tensor decomposition by using coordinate-based MLP, which may have a potential impact on future research.

---

### Official Review · Reviewer_MGAi · 2021-11-02

**Correctness:** 4
**Technical Novelty And Significance:** 4
**Empirical Novelty And Significance:** 3
**Recommendation:** 8
**Confidence:** 4

**Main Review:**

My take:
When I first looked over this paper, I thought to myself, “That’s crazy! Does that actually work?” When I read it in detail, I thought, “Okay, the authors have definitely done their due diligence, and thought through a lot of the caveats needed to get this idea to work and to appropriately reap the benefits.” I then took to implementing it myself in a range of benchmark tasks involving implicit neural representations and in other architectures. As it turns out, this absolutely works as advertised, is extremely easy to implement, and provides the advertised compute speedups, albeit at a slight cost in parametric efficiency. What’s more, some digging turns up that there’s actually a reasonably solid theoretical justification for this approach: as noted by Hinton in arxiv.org/abs/2102.12627, “The Kolmogorov-Arnold superposition theorem states that every multivariate continuous function can be represented as a superposition of continuous functions of one variable.” I think this paper needs a bit of cleanup on the presentation and clarity side, but is clearly in accept territory. I am willing to champion this paper: while I don’t necessarily think its impact potential is massive, I think it does have the potential to have a reasonable degree of impact, and I especially believe that we should highlight solid work which teaches us something *surprising*.


Detailed notes:
- The authors mention that they require more parameters to reach a given performance level a few times, but I think they should be more upfront in specifying that this is a drawback of the method. I have experimented extensively with this and it is quite clear that, if one keeps a fixed reconstruction accuracy target, CoordX models require slightly more parameters than a non-CoordX model to reach that target regardless of architecture; this can also be seen in lines 1 and 3 of Table 5. As most readers/practitioners likely won’t care about a reduction in parametric efficiency due to the improvements in compute efficiency, I don’t think this will make the paper seem any weaker (and is more honest anyway). Specifically I would like to see mentioned in the second bullet point under contributions that the parameter count is *comparable* but is almost necessarily *slightly larger.* This also retains the specific design of the weight-sharing layers as a visible contribution.


-One “drawback” of this model is that it does not improve things in the unidimensional coordinate case. This is not actually a drawback, but rather a lack of a benefit, but that’s perfectly fine. It might be worth mentioning this as e.g. audio representation would not benefit here.

Minor:
-The very first sentence of the introduction appears to be some sort of typo or accidentally left-in mistake (“recently coordx…”). Please fix this.


**Summary Of The Paper:**

Summary: This paper proposes a modification to INR models on multidimensional coordinate grids where a subset of the earlier layers operate on the decomposed coordinate grid. In this setup, the grid (which is assumed to be regularly sampled to permit this decomposition) is broken into its constituent components, (e.g. x and y instead of (x,y)), passed through a single linear layer unique to that component, then through a stack of shared layers, followed by an outer product to return to the joint (x,y) space, and at least one layer that operates on the joint space. This approach lightly reduces parametric efficiency but strongly improves compute efficiency (both in terms of FLOPS, memory usage, and actual observed runtime) for common implicit neural representation tasks, including fitting images, videos, shapes, and volumetric rendering via radiance fields.


**Summary Of The Review:**

This paper presents a technique of general interest to the subfield of implicit neural representations, has solid empirical results, and is in my opinion quite surprising. The paper is clearly in accept territory.

---

> ### Author Response · Authors · 2021-11-16
> **Response to Reviewer MGAi**
>
> Thanks for the positive assessment and insightful comments. We also appreciate your great efforts in implementing and verifying our method. Below we respond to the points mentioned in the review.
>
> **Parametric efficiency.** We agree that the drop in the parametric efficiency is a drawback of our method. We have clarified this under the contributions in the Introduction section of our revised submission.
>
> **Unidimensional signal.** Thank you for the feedback. We have mentioned this in the revised submission (as a footnote on Page 2).
>
> **Minor writing issue.** Thank you for pointing this out. We have fixed this writing issue in the revised submission.
>
> **Other comments.** We thank the reviewer for the feedback -- we believe that our method may have a potential theoretic impact, beyond the scope of neural implicit representations acceleration,  for example, using the CoordX method to perform meaningful matrix decompositions or multivariate function decomposition [1] as mentioned by the reviewer (we appreciate the pointer to the work). We intend to explore this direction in future work.
>
> [1] How to represent part-whole hierarchies in a neural network, Hinton.

---

### Author Response · Authors · 2021-11-16
**Summary of Paper Revision**

We thank the reviewers for their constructive feedback. We have uploaded the revised paper that addresses the reviewers’ comments. To summarize, we have added the following:
1. Fixed the statement “a slight drop in quality” under Section 4.1, we now corrected the statement to reflect the ~3dB drop in quality as mentioned by reviewer z7hu.
2. Fixed the equation notation issues mentioned by reviewer z7hu for equation (2).
3. Rewrote the first paragraph in the introduction and removed the misplaced sentence.
4. Added a citation to [1] under Section 5, where we mentioned and suggested future exploration on data compression using our CoordX architecture.
5. Added a short section to discuss the NeRF models in the main text (last part of Section 4.1) as suggested by reviewer xWSd.
6. Included more details for input point sampling under Section 3.2, according to reviewer xWSd’s feedback.
7. Added an ablation on feature fusion comparing different fusion strategies (multiplicative modulation, additive modulation, and concatenation) under section B.4 of appendix B, according to reviewer xWSd’s feedback.
8. Modified the second bullet point under contribution in the introduction section, instead of saying “model size unchanged”, we now mention that the parameter count is comparable or slightly larger, as suggested by reviewer MGAi.
9. Mentioned that unidimensional signal fitting (e.g., audio representation) tasks do not benefit from CoordX under Section 1 as suggested by reviewer MGAi.
10. Added two more NeRF experiments (Ship & Drums) to Appendix C.
11. Added a discussion of using CoordX for NeRF and the evaluation results in the main text under Section 4 as suggested by reviewer xWSd.
12. Added a more detailed discussion of ACORN [1] to Section 2 as mentioned by reviewer z7hu.
13. Thoroughly proofread the paper and fixed various typos and grammatical errors.

We thank the reviewers again for their feedback. We hope our modifications and clarifications have addressed your concerns about our paper. We would be happy to discuss and answer any other questions you may have.

[1] COIN: COmpression with Implicit Neural representations, Dupont et al.

[2] ACORN: Adaptive Coordinate Networks for Neural Scene Representation, Martel et al.

---

### Decision · Program_Chairs · 2022-01-20

**Decision:**

Accept (Poster)

**Comment:**

Implicit neural representations are a new and promising method to represent images and scenes. Implicit neural representations enable good performance on task like view synthesis. Those networks generate an image of scene pixel-by-pixel and are therefore computationally expensive. The paper proposes a method to accelerate inference with an MLP by learning each dimension of the input (e.g., x, y, and z coordinates) separately. The paper reports speedups by a factor of up to three.

The four reviewers all agree that this paper should be accepted. The reviewer's highlight that the speedup is a solid technical contribution, and agree that the idea of splitting the coordinates is well motivated and well evaluated, and leads to the advertised speedups.
During the review process, the reviewers also raised some issues, such as a slight performance loss at that cost of a slight increase in efficiency, but were convinced by the response of the reviewers.

I recommend accepting the paper. Like the reviewers, I think that the proposed idea is interesting and technically sound, however, I'm a bit concerned about the slight drop in performance: If a slight drop of performance is allowed, then a speedup is possible by simply making the networks smaller, reducing the layers, or through other more immediate means than the proposed one. The contribution would be even more convincing if the paper also compares the speedup in a fair setup where the performance is kept constant.